# Lipid-induced transcriptomic changes in blood link to lipid metabolism and allergic response

Koen F. Dekkers[1], Roderick C. Slieker [2,3], Andreea Ioan-Facsinay[4], Maarten van Iterson[1], BIOS consortium*, M. Arfan Ikram [5], Marleen M. J. van Greevenbroek [6], Jan H. Veldink [7], Lude Franke [8], Dorret I. Boomsma [9], P. Eline Slagboom [1], J. Wouter Jukema [10,11] & Bastiaan T. Heijmans [1] ✉

Immune cell function can be altered by lipids in circulation, a process potentially relevant to lipid-associated inflammatory diseases including atherosclerosis and rheumatoid arthritis. To gain further insight in the molecular changes involved, we here perform a transcriptome-wide association analysis of blood triglycerides, HDL cholesterol, and LDL cholesterol in 3229 individuals, followed by a systematic bidirectional Mendelian randomization analysis to assess the direction of effects and control for pleiotropy. Triglycerides are found to induce transcriptional changes in 55 genes and HDL cholesterol in 5 genes. The function and cell-specific expression pattern of these genes implies that triglycerides downregulate both cellular lipid metabolism and, unexpectedly, allergic response. Indeed, a Mendelian randomization approach based on GWAS summary statistics indicates that several of these genes, including interleukin-4 (*IL4*) and IgE receptors (*FCER1A*, *MS4A2*), affect the incidence of allergic diseases. Our findings highlight the interplay between triglycerides and immune cells in allergic disease.

Immune cells are continuously exposed to external stimuli that can lead to phenotypic[1] and molecular[2] changes. Recent studies have shown that blood lipids, including cholesterol[3] and triglycerides[4], can affect circulating immune cells. This process may be relevant for lipid-associated inflammatory diseases such as atherosclerosis[5] and rheumatoid arthritis[6]. However, the effect of blood lipids on transcription, a process particularly informative of the molecular state of a circulating immune cell[7], is largely unknown.

Several studies have investigated the relationship between lipids and immune cells. Tissue culture experiments, for example, indicated that exposure to lipids can affect the transcription of genes in immune cells in vitro[3,8,9]. An increasingly recognized and powerful alternative

[1]Molecular Epidemiology, Department of Biomedical Data Sciences, Leiden University Medical Center, 2333 ZC Leiden, The Netherlands. [2]Department of Cell and Chemical Biology, Leiden University Medical Center, 2333 ZC Leiden, The Netherlands. [3]Department of Epidemiology and Data Science, Amsterdam UMC, VUmc, 1081 HV Amsterdam, The Netherlands. [4]Department of Rheumatology, Leiden University Medical Center, 2333 ZA Leiden, The Netherlands. [5]Department of Epidemiology, Erasmus MC, 2015 GD Rotterdam, The Netherlands. [6]Department of Internal Medicine and CARIM School for Cardiovascular Diseases, Maastricht University Medical Center, 6229 HX Maastricht, The Netherlands. [7]Department of Neurology, University Medical Center Utrecht Brain Center, Utrecht University, 3584 CX Utrecht, The Netherlands. [8]Department of Genetics, University Medical Center Groningen, 9713 GZ Groningen, The Netherlands. [9]Department of Biological Psychology, Vrije Universiteit Amsterdam, 1081 HV Amsterdam, The Netherlands. [10]Department of Cardiology, Leiden University Medical Center, 2333 ZA Leiden, The Netherlands. [11]Netherlands Heart Institute, 3501 DG Utrecht, The Netherlands. *A list of authors and their affiliations appears at the end of the paper. ✉e-mail: b.t.heijmans@lumc.nl

approach to infer such relationships in vivo is the use of population genomics data in combination with Mendelian randomization analysis (MR). MR uses genetic variants as causal anchors to infer directed relationships[10,11]. Using this approach, we[4] and others[12] observed an effect of lipids on DNA methylation, an epigenetic mark, which was linked to end-product feedback control of lipid metabolism. Similarly, blood lipid levels have been reported to be associated with differential expression in blood. Although the smaller scale of the latter studies rendered an MR analysis to infer the direction of effects unsuccessful, they did suggest that lipid-associated differential expression is more widespread than differential DNA methylation[13,14].

Therefore, we adopted a two-step approach to determine whether lipids influence transcription in circulating immune cells. First, we performed a large-scale transcriptome-wide analysis of blood triglycerides (TG), HDL cholesterol (HDL-C), and LDL cholesterol (LDL-C) in whole blood samples of 3229 individuals. Next, building on our previous work[4], we implemented a comprehensive MR analysis to infer causal effects of lipid levels on transcription using multiple genetic variants as causal anchors that allowed us to systematically investigate potential pleiotropy using state-of-the-art methods[15,16]. Our analysis revealed that particularly TG affects the blood transcriptome by downregulating genes involved in cellular lipid metabolism and the allergic response.

## Results

### Transcriptome-wide analysis reveals genes associated with lipid levels

To evaluate the association of blood lipids with transcription in circulating immune cells, we performed a transcriptome-wide analysis in 3229 individuals with whole blood RNA-seq data from 6 cohorts participating in the BIOS consortium (Table 1). We identified 496 differentially expressed genes for TG, 384 for HDL-C, and 26 for LDL-C ($P$-value $< 2.8 \times 10^{-6}$; Supplementary Data 1). The associations observed for TG not only stood out in number but also in strength in terms of effect size and $P$-value (Fig. 1a). In line with correlations between lipid levels (in our study: $R_{TG-HDL-C} = -0.48$, $R_{TG-LDL-C} = 0.45$, and $R_{HDL-C-LDL-C} = -0.12$), there was substantial overlap in genes. We observed 194, 195, and 23 genes for TG-C, HDL-C, and LDL-C, respectively, that were associated with at least one additional other lipid level (Fig. 1b). Evidence for the associations was generally consistent across cohorts: for 757 of 906 associations the direction was consistent for all 6 cohorts, for 143 associations the direction was consistent for 5 of 6 cohorts, for 5 associations the direction was consistent for 4 of 6 cohorts, and for 1 association the direction was consistent for 3 of 6 cohorts (Supplementary Data 1). Effect sizes were not sensitive to additional adjustments for smoking behavior or lipid-lowering medication (Supplementary Fig. 1).

### Constructed genetic instruments are valid instruments for Mendelian randomization

To infer causal relationships using MR, we first constructed weighted genetic instrumental variables (GIVs) for blood lipids from genetic variants reported in a genome-wide association study of lipid levels among 188,577 individuals[17] (TG: 40 variants, HDL-C: 69 variants, LDL-C: 57 variants; all variants were available in the current study; Supplementary Data 2). The GIVs were strongly associated with their respective lipid levels in our own study ($F$-stat $> 134$, $P$-value $< 10^{-31}$; Supplementary Data 3), although they explained a minor proportion of the total variance ($R^2 = 4.0-6.4\%$). There was an overlap between the variants of the lipid GIVs. Of the variants, 18/40, 49/69, and 44/57 variants were unique for TG, HDL-C, and LDL-C, respectively. The GIVs were not associated with measured potential confounders' age, sex, and cell counts, a necessity for accurate effect estimation, except for the LDL-C GIV, which was nominally associated with monocyte fraction in whole blood ($P$-value $= 0.011$; Supplementary Data 4). This association was weak, however, when compared with the strong association of the LDL-C GIV and LDL-C levels ($P$-value $= 2.6 \times 10^{-31}$). Together, this indicates that our GIVs are valid instruments for MR.

### Mendelian randomization reveals genes affected by lipid levels

To infer a causal effect of lipids on transcription using the GIVs as proxies for lipid levels, we performed an MR analysis for the 906 associations found in the transcriptome-wide analysis. We used a modified Cochran's $Q$-test[15] to account for pleiotropy by iteratively removing genetic variants from the GIVs until no pleiotropy was detected ($P_Q > 0.05$). We found evidence of an effect of TG on 56 genes, of HDL-C on 6 genes, and of LDL-C on 0 genes after adjustment for multiple testing using the Benjamini–Hochberg method at 5% false discovery rate ($P_{FDR} < 0.05$, Supplementary Data 5). Pleiotropic variants were removed from the GIVs for 11 of the TG effects and 5 of the HDL-C effects (Supplementary Data 5). MR effect size estimates were concordant with those previously observed in the transcriptome-wide analysis (Fig. 2a). Most of the effects for TG and LDL-C were negatively correlated with transcription (TG: 52 of 56), while most of the effects for HDL-C were positively correlated (5 of 6). A post-hoc power analysis indicated that our MR analysis was able to identify only ~6% of effects with 80% power; we detected ~7% (62 of 906 associations; Supplementary Data 5, Supplementary Fig. 2). In line with the observed overlap between genes associated with different lipid levels, all 6 genes for which we found evidence for an effect of HDL-C on transcription were also observed for TG (Fig. 2b).

We also extended the MR analysis to evaluate the opposite direction of effect, i.e. whether transcription of the 62 genes observed in the MR analysis can affect lipid levels, a procedure known as

**Table 1 | Characteristics of the six cohorts in the BIOS consortium**

| | CODAM | LL | LLS | NTR | PAN | RS |
|---|---|---|---|---|---|---|
| $N$ | 183 | 741 | 642 | 797 | 169 | 697 |
| Sex (% Male) | 56 | 42 | 47 | 35 | 63 | 42 |
| Age (years, SD) | 65.33 (7.05) | 45.33 (13.15) | 58.83 (6.61) | 38.18 (15.21) | 62.55 (9.55) | 67.61 (5.93) |
| TG (mmol/L, SD) | 1.58 (0.8) | 1.14 (0.88) | 1.93 (1.22) | 1.3 (0.74) | 1.85 (1.07) | 1.48 (0.86) |
| LDL-C (mmol/L, SD) | 3.93 (1) | 3.3 (0.97) | 3.82 (0.98) | 3.2 (0.96) | 3.83 (0.97) | 3.71 (0.94) |
| HDL-C (mmol/L, SD) | 1.34 (0.33) | 1.55 (0.41) | 1.42 (0.44) | 1.43 (0.38) | 1.44 (0.37) | 1.51 (0.44) |
| Monocytes (%, SD) | 7.13 (2.11) | 7.47 (2.42) | 7.35 (2.43) | 7.96 (2.8) | 7.28 (2.46) | 7.46 (2.41) |
| Lymphocytes (%, SD) | 32.48 (10.23) | 31.87 (8.56) | 32.69 (8.16) | 34.59 (8.68) | 32.58 (8.6) | 33.93 (7.9) |
| Neutrophils (%, SD) | 57.12 (10.99) | 56.81 (9.39) | 55.98 (8.72) | 53.8 (9.39) | 56.44 (8.88) | 54.65 (8.6) |
| Eosinophils (%, SD) | 2.75 (2.03) | 2.89 (1.85) | 2.79 (1.63) | 3.04 (1.99) | 2.82 (1.73) | 2.91 (1.82) |
| WBC (×10⁹ cells/L, SD) | 6.76 (1.52) | 6.52 (1.41) | 6.61 (1.63) | 6.41 (1.57) | 6.95 (2.06) | 6.44 (1.7) |
| RBC (×10¹² cells/L, SD) | 4.77 (0.37) | 4.71 (0.34) | 4.72 (0.35) | 4.71 (0.39) | 4.66 (0.34) | 4.68 (0.36) |

*CODAM* Cohort on Diabetes and Atherosclerosis Maastricht, *LL* LifeLines, *LLS* Leiden Longevity Study, *NTR* Netherlands Twin Register, *PAN* Prospective ALS Study Netherlands, *RS* Rotterdam Study, *SD* standard deviation.

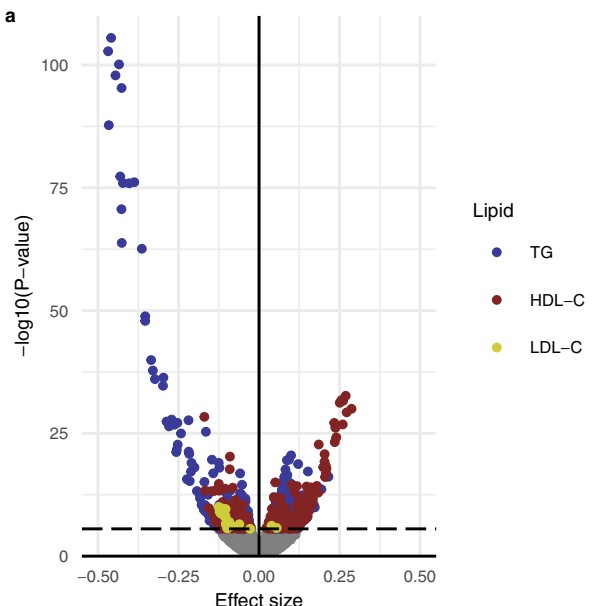

**Fig. 1 | A transcriptome-wide analysis reveals genes associated with lipids.**
**a** Volcano plot depicting the relationship between linear regression effect size in the standard deviation of lipid levels and −log10(*P*-value) for the association between lipid levels and transcription for TG, HDL-C, and LDL. Points depicted in color represent genome-wide significant associations. The dashed horizontal line represents the Bonferroni threshold based on 17,740 tests, i.e. the number of genes. **b** Overlap between genes associated with lipid levels for TG, HDL-C, and LDL-C.

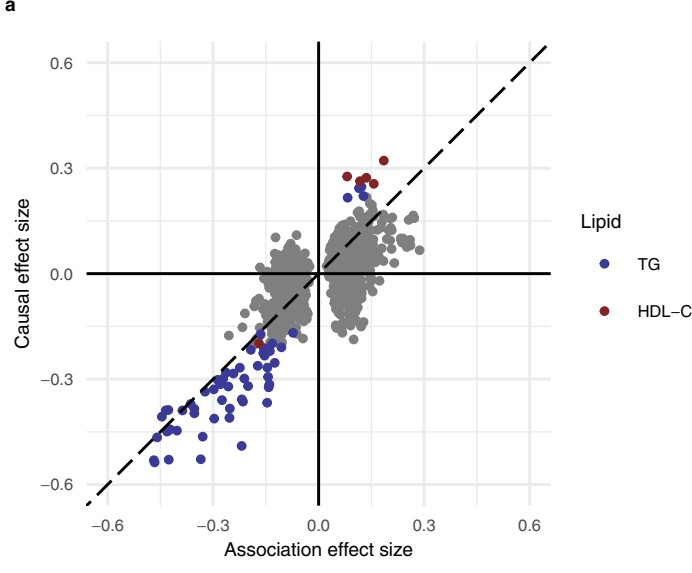

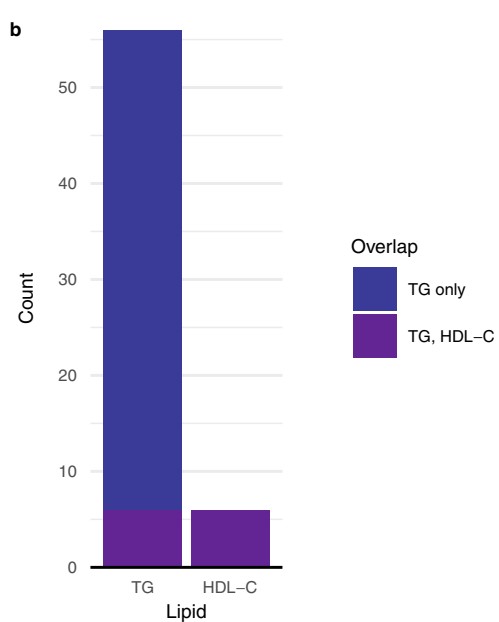

**Fig. 2 | Mendelian randomization reveals genes affected by lipids. a** Comparison between association (transcriptome-wide analysis) linear regression and causal (MR) Wald ratio effect sizes for TG, HDL-C, and LDL-C. The point depicted in color represents FDR-significant effects. **b** Overlap between genes affected by lipid levels for TG, HDL-C, and LDL-C.

bidirectional MR[11]. Using the strongest associating expression quantitative trait locus (QTL) for each gene previously identified[18] as a proxy for transcription, we found no evidence of reverse causation ($P_{FDR} \geq 0.15$; Supplementary Data 6).

**Systematic sensitivity analysis of pleiotropy**
We investigated whether Cochran's method missed residual pleiotropy, which would violate the MR assumption that the effect is mediated by lipid levels. An important source of pleiotropy is the case where GIV variants are located near genes identified in the transcriptome-wide

analysis and directly influence the expression through an expression QTL effect *in cis*. As we previously showed, correction for this source of direct pleiotropy can be achieved by including the variant as a covariate in the MR model[4]. This adjustment corroborated the results of Cochran's method for all identified effects and both adjustments negated the case of pleiotropy in which the variant rs174546 has a direct effect on LPL expression not mediated by TG level (*P*-value < 0.05; Supplementary Fig. 3).

Egger regression, a previous alternative to Cochran's *Q*-test to test for pleiotropy[19], indicated the presence of residual directional

pleiotropy for 3 out of 62 genes (*SLC27A2* (TG), *LYBD6B* (TG) and *AC004381.6* (TG); Supplementary Fig. 4) but only for *LYBD6B* did Egger regression appreciably alter the effect size (*P*-value < 0.05).

Finally, effect sizes were generally not sensitive to further adjustment for GIVs for the other lipids using a multivariable MR analysis[20]. This is important because there was an overlap between genetic variants in the lipid GIVs. After adjustment for the effect of the lipid of interest for the GIVs of the other lipids, all associations were confirmed except for *CCNA1*, which was no longer affected by HDL-C after adjustment for the TG GIV (Supplementary Fig. 4). Similarly, we constructed GIVs for other potential pleiotropic factors, namely systolic blood pressure, diastolic blood pressure, and BMI, using public GWAS data[21,22] and subsequently applied the same procedure as implemented for the lipid GIVs (Supplementary Fig. 4). This analysis did not detect any further pleiotropic associations.

In summary, our systematic analysis of pleiotropy indicates that Cochran's method was generally successful in detecting and correcting for pleiotropy with the exception of 2 out of 62 effects (*LYBD6B* for TG and *CCNA1* for HDL-C). This resulted in a final set of 55 genes whose expression was influenced by TG and 5 genes whose expression was influenced by HDL-C (Supplementary Data 5). As noted previously, HDL-C-influenced genes were also identified as TG-influenced genes. Our extensive analysis did not indicate pleiotropy and thus favors the interpretation that the effects on TG and HDL-C were independent.

The majority of the genes we identified constitute novel findings, but there also was considerable overlap with earlier findings thus corroborating the results. A recent study implementing a two-sample MR method reported 29 genes putatively affected by TG and/or HDL-C[23] and our set of 55 genes included 17 of those genes (*P*-value < $10^{-10}$) (Supplementary Data 7). Similarly, part of the genes identified in our MR analysis were reported in previous transcriptome-wide association analyses[13,14], namely 20 of the 55 genes affected by TG and 3 of the 5 genes affected by HDL-C (Supplementary Data 7). Notably, for a previously reported 'lipid-leukocyte' co-expression module[14], we now show that the differential transcription observed was induced by TG for 9 out of the 11 genes in that module.

## Genes affected by lipids are enriched for lipid metabolism and allergy

To gain insight into the biological processes that are differentially regulated in immune cells by TG, we performed an enrichment analysis for the 55 genes whose expression was affected by TG using 10 human pathway databases. We detected 27 enriched processes primarily defined by two subsets of TG-affected genes ($P_{FDR}$ < 0.05; Fig. 3a, Supplementary Data 8).

Seven processes were related to lipid metabolism and included central regulators of this overarching process (*ABCA1, SQLE, HPGDS, CYP11A1, ACSL6, SLC27A2, SREBF2, ABCG1,* and *PLD3*). The TG-affected genes also included two additional genes known to be involved in lipid metabolism, but not reported in the pathway databases: *CAV2*, a component of lipid rafts; *MYLIP*, a sterol-dependent inhibitor of cellular cholesterol uptake that mediates ubiquitination and subsequent degradation of LDLR. The expression of 6 of the 9 lipid-related genes in blood cells was down-regulated in the presence of high plasma TG (Fig. 2a; Supplementary Data 5). Notably, all genes affected by HDL-C are part of this lipid metabolism subset.

However, 12 processes, including the top-9 enriched processes, were related to allergy. They encompassed 10 TG-affected genes and included factors in the signal transduction mediating the allergic response, like interleukin-4 (*IL4*), IgE receptors (*FCER1A, MS4A2*), various other receptors (*IL1RL1, HRH4, CCR3,* and *PTGER3*), and enzymes (*HDC, HPGDS,* and *CYP11A1*). The genes *HPGDS* and *CYP11A1* were part of both lipid metabolism and allergy subsets, in line with their function in prostaglandin metabolism. A role in allergy extended beyond the TG-affected genes reported in the pathway databases, namely *RP11-*

*13A1.1* (a lncRNA implicated in fungal immune response) and *CPA3* (a protease released by mast cells and basophils, whose expression is elevated in asthma patients). Elevated TG levels were indicated to uniformly downregulate the expression of these genes (Fig. 2a; Supplementary Data 5).

The TG-affected genes showed a cell-specific gene expression pattern that was concordant with gene function. An analysis of public RNA-seq data[24] indicated that TG-affected genes involved in lipid metabolism were ubiquitously expressed across all cell types. In contrast, the genes involved in allergy were expressed at low levels in all cell types except in PBMCs, progenitor cells, and, most prominently, in basophils, a cell type that is intimately involved in allergic reactions (Fig. 3b).

To evaluate the relevance of these findings, we investigated whether the TG-affected genes were also linked to allergy-related phenotypes through a two-sample MR analysis. We retrieved summary statistics for genome-wide association studies of 4 allergic phenotypes[25–27] and *cis*-expression QTLs[18] for TG-affected genes as proxies for transcription. We observed an effect of *IL4* (*P*-value = $2.7 \times 10^{-7}$), *IL1RL1* (*P*-value = $1.3 \times 10^{-3}$), *RUNX1* (*P*-value = $6.3 \times 10^{-5}$), *FCER1A* (*P*-value = $3.1 \times 10^{-4}$), and *RP11-13A1.1* (*P*-value = $2.9 \times 10^{-3}$) on the incidence of a combined allergic disease phenotype (asthma and/or hay fever and/or eczema)[26], an effect of *IL4* (*P*-value = $1.0 \times 10^{-13}$), *IL1RL1* (*P*-value = $4.7 \times 10^{-4}$), *RUNX1* (*P*-value = $1.2 \times 10^{-4}$) and *RP11-13A1.1* (*P*-value = $1.5 \times 10^{-3}$) on the incidence of childhood-onset asthma[25], an effect of *IL4* (*P*-value = $2.6 \times 10^{-4}$), *FCER1A* (*P*-value = $1.4 \times 10^{-5}$), *RUNX1* (*P*-value = $3.4 \times 10^{-6}$), *ACSL6* (*P*-value = $2.4 \times 10^{-3}$), and *MS4A2* (*P*-value = $5.8 \times 10^{-4}$) on the incidence of adult-onset asthma[25] and an effect of *FCER1A* (*P*-value = $3.9 \times 10^{-11}$) on IgE levels[27]. Notably, the directionality of the effects was consistent across phenotypic outcomes (Supplementary Data 9).

Finally, we performed the same two-sample MR analysis for atherosclerosis-related outcomes and rheumatoid arthritis, two lipid-associated inflammatory diseases for which we a priori hypothesized that lipid-induced changes in the transcriptome of immune cells could be relevant. Based on summary statistics from GWASs of coronary artery disease[28], myocardial infarction[28], and rheumatoid arthritis[29], only for *CCR3* (*P*-value = $6.0 \times 10^{-5}$), an effect on rheumatoid arthritis was indicated (Supplementary Data 9). Several other TG-affected genes showed weaker associations with atherosclerosis-related outcomes (*ABCG1, AC004791.2, CPA3, CYP11A1, SLC12A3*) or rheumatoid arthritis (*GATA2, SMPDL3A*) but were no longer statistically significant after correction for multiple testing ($P_{FDR}$ > 0.05; Supplementary Data 9).

## Discussion

We performed a transcriptome-wide analysis in the blood cells of 3229 individuals and identified 496 differentially expressed genes for TG, 384 for HDL-C, and 26 for LDL-C. We then performed an MR analysis and identified 55 genes affected by TG (of which 35 have not been identified in previous transcriptome-wide analyses) and 5 genes affected by HDL-C (of which 2 have not been identified in previous transcriptome-wide analyses). The genes affected by TG were enriched for lipid metabolism and allergy pathways. Moreover, the TG-affected genes involved in allergy pathways were specifically expressed in basophils and were linked to allergy-related outcomes using a two-sample MR analysis.

All of the five genes affected by HDL-C were also affected by TG and are central players in lipid metabolism. In line with the negative correlation between blood HDL-C and TG levels, all the affected genes had the opposite direction of effect for TG and HDL-C. Our systematic analysis of pleiotropy indicated that the TG and HDL-C effects are independent. Although such analyses are unable to definitely prove whether this is the actual mechanistic route in vivo, there is evidence for 2 of the 5 overlapping genes that both effects do occur simultaneously. Mutations in *ABCA1* lead to increased plasma TG levels and

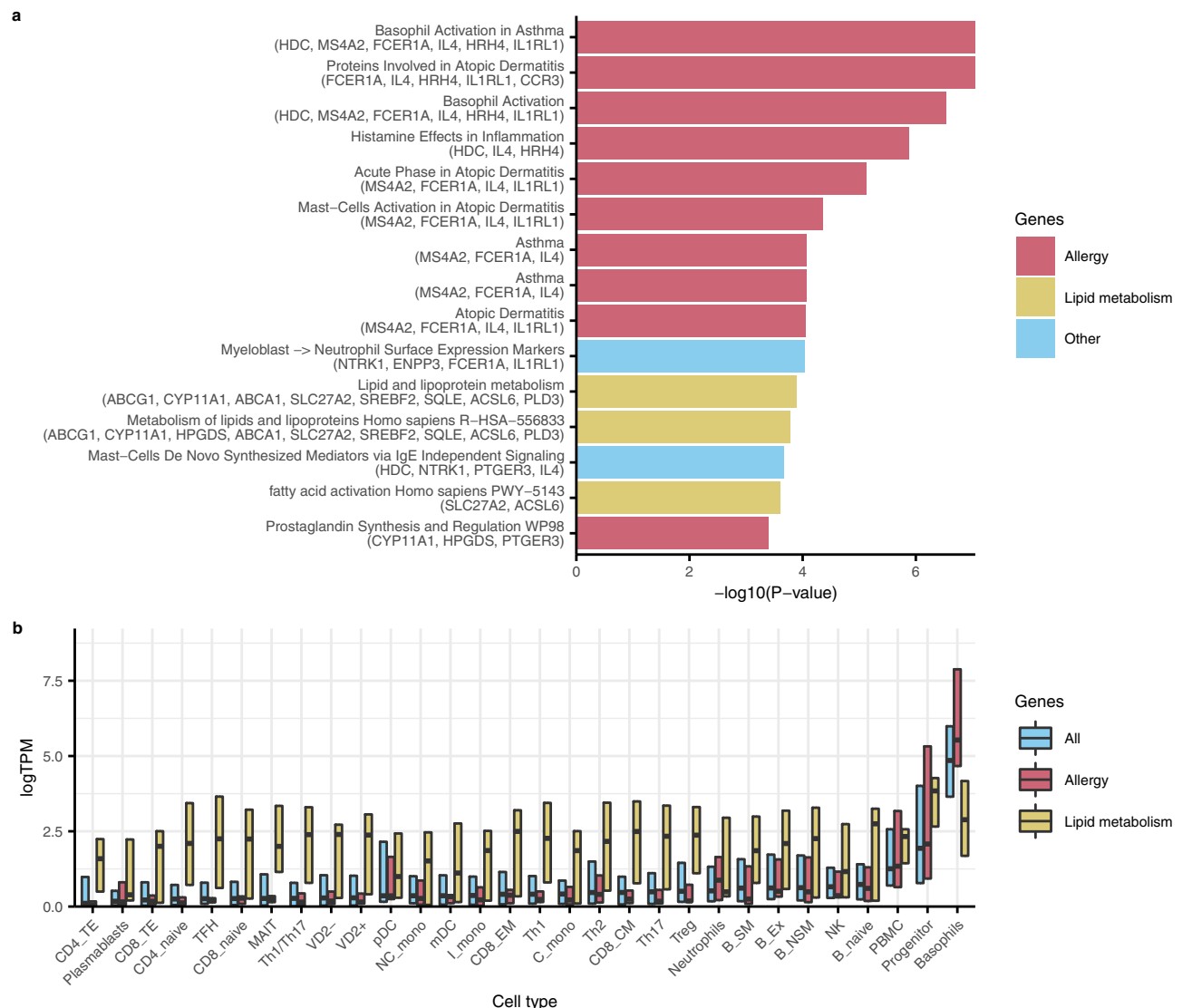

**Fig. 3 | TG-affected genes are enriched for lipid metabolism and allergy pathways and show cell-specific expression patterns in line with gene function. a** Pathway enrichments generated using a one-sided Fisher's exact test with clusterProfiler[56] using the 10 human pathway databases BioPlanet 2019, WikiPathways 2019 Human, KEGG 2019 Human, Elsevier Pathway Collection, BioCarta 2015, Reactome 2016, HumanCyc 2016, NCI-Nature 2016, Panther 2016, and MSigDB Hallmark 2020. The top 15 enrichments are shown. **b** Lipid metabolism genes are expressed in all blood cell types, but genes involved in allergy are expressed especially in basophils. Depicted are medians and interquartile range. Based on public RNA-seq data[24] of T follicular helper (TFH), T regulatory cells (Treg), T helper 1 (Th1), T helper 1/T helper 17 (Th1.Th17), T helper 17 (Th17), T helper 2 (Th2), T CD4 terminal effector (CD4_TE), T CD4 Naive (CD4_naive), Υ/δ Vd2+ (VD2.), Υ/δ Vd2- (VD2..1), MAIT, T CD8 Naive (CD8_naive), T CD8 central memory (CD8_CM), T CD8 effector memory (CD8_EM), T CD8 terminal effector (CD8_TE), Progenitor cells, Naive B cells (B_naive), Non-switched memory (NSM) B cells (B_NSM), Exausted (Ex) B cells (B_Ex), Switched memory (SM) B cells (B_SM), Plasmablasts, Low-density (LD) neutrophils, NK cells (NK), Classical (C) monocytes (C_mono), Intermediate (I) monocytes (I_mono), Non-classical (NC) monocytes (NC_mono), Myeloid Dendritic Cells (mDC), Plasmacytoid Dendritic Cells (pDC), low-density (LD) basophils, and peripheral blood mononuclear cells (PBMC).

decreased HDL-C levels in patients[30], while *ABCG1*, in addition to its role in cholesterol efflux, also promotes lipid accumulation in the presence of triglyceride-rich lipoproteins[31]. According to our analysis, higher TG leads to decreased transcription of both genes, while higher HDL-C leads to increased transcription, which is consistent with end-product feedback control. This process, where the end product inhibits its own synthesis and which has been observed for cholesterol[32], is what we proposed as an explanation for the results of our previous study, in which we showed that lipids affect DNA methylation of genes involved in lipid metabolism[4].

Multiple genes that were affected by TG have a well-established role in allergy. An allergic response is initiated by T-helper 2 cells that secrete IL4 in response to an allergen, which in turn promotes B cells to produce IgE. IgE binds to the IgE receptor on basophils and mast cells

which start secreting inflammatory mediators, such as cytokines, histamine, proteases, and prostaglandins, a group of lipids with hormone-like effects. The TG-affected genes included two IgE receptor genes, *FCER1A* and *MS4A2*, the allergic response initiating cytokine *IL4*, two genes involved in histamine metabolism, *HDC* and *HRH4*, a protease released by mast cells and basophils, *CPA3*, and three genes involved in prostaglandin metabolism, *HPGDS, CYP11A1*, and *PTGER3*. All the genes were indicated to be downregulated in blood cells by elevated TG levels. Although some of the genes are specific to the allergic response, most of them have multiple immune functions and it is not certain that TG affects this pathway specifically.

Our MR analysis had sufficient statistical power to find putatively causal relationships between lipid levels and the subset of genes that we first identified in a transcriptome-wide association analysis.

We used Cochran's method[15] which was recently adapted for use in MR to account for pleiotropy and subsequently performed a series of sensitivity analyses that supported the validity of our results. Despite detecting causal relationships between lipid levels and transcription in blood, a post-hoc power analysis also revealed that our MR analysis will have missed a substantial proportion of effects; for example, a study size larger than a million individuals is required to have sufficient statistical power to detect 90% of the effects. While post-hoc power calculations are not a valid method to estimate the statistical power of a study[33], it does provide a reasonable estimate of the order of magnitude of the sample sizes that MR studies require.

Limitations of our analysis include that it was based on whole blood and that information about allergy was unavailable in our own data. However, an analysis of public data[24], showed that lipid metabolism genes were expressed in all cell types, whereas genes involved in allergy were strongly expressed specifically in basophils. Basophils play an important role in immune responses, including allergic response[34]. Furthermore, a two-sample MR approach leveraging large GWAS and eQTL databases suggested a potential causal role for *IL4*, *IL1RL1*, *RP11-13A1.1*, *FCER1A* and *MS4A2, RUNX1*, and *ACLS6* expression in various allergic diseases and serum IgE levels. However, since this analysis was based on a single genetic variant (the SNP responsible for the strongest eQTL per gene of interest), we could not perform the sensitivity analyses that we used for the previous MR analysis to rule out pleiotropy. Also, eQTL effects were not specific for basophils but reported for whole blood and, although detected in whole blood, effects may extend to cell types outside the circulation. Further studies are needed to elucidate whether lipids can actually affect cells in a cell type-specific way and how they might affect the allergic response.

Our findings highlight the interplay between blood TG and circulating immune cells in the allergy response, a largely unappreciated phenomenon. Our original hypothesis, however, was that the interaction between blood lipids and immune cells was particularly relevant for lipid-associated inflammatory diseases including atherosclerosis[5] and rheumatoid arthritis[6]. With the exception of the *CCR3* gene for rheumatoid arthritis, none of the TG-affected genes were associated with these diseases on the basis of the two-sample MR approach. This negative result should be interpreted with caution in view of the limitations of the approach, including a low statistical power, and the observation that some genes did show an association before the necessary correction for multiple testing. While any contribution is less apparent than for the allergic response, other study designs will be required to obtain a definite answer.

In conclusion, our two-step analysis approach showcases the potential of transcriptome-wide analyses followed by MR when combined with rigorous testing of assumptions regarding pleiotropy. Application of this approach to large-scale multiple omics studies resulted in many genes previously unknown to be affected by lipid levels in blood and suggested a novel, causal role for triglycerides in the downregulation of the allergic response.

# Methods

## Cohorts
The Biobank-based Integrative Omics Study (BIOS) Consortium[35,36] comprises six Dutch cohorts: Cohort on Diabetes and Atherosclerosis Maastricht (CODAM)[37], LifeLines (LL, https://www.lifelines.nl/)[38], Leiden Longevity Study (LLS, https://leidenlangleven.nl/)[39], the Netherlands Twin Register (NTR, https://tweelingenregister.vu.nl/)[40], Rotterdam Study (RS, https://www.ergo-onderzoek.nl/)[41], and Prospective ALS Study Netherlands (PAN, https://www.als-centrum.nl/kennisplatform/biobank-neuromusculaire-ziekten-nmz/)[42]. The study was approved by the institutional review boards of the participating centers and all participants have given written informed consent and the experimental methods comply with the Helsinki Declaration. Genotype, RNA-seq, and blood profiles (including lipid levels and cell counts) were available in whole blood, which was collected simultaneously for all measurements. The measurements of the samples for the genetics, lipids and cell counts data were performed individually by the cohorts. All RNA-seq data were generated centrally within the BIOS consortium by the Human Genotyping facility (HugeF) of ErasmusMC, the Netherlands (http://www.glimdna.org/). Characteristics of the cohorts can be found in Table 1.

## Lipids
Triglyceride (TG), HDL cholesterol (HDL-C), and total cholesterol levels (TC) were measured after a fasting period of 12 h for CODAM, LL, NTR, RS, and PAN; for LLS non-fasted lipids were measured. LDL cholesterol (LDL-C) was calculated using Friedewald's method[43]. To address the non-normality of the distributions of lipid levels and to be consistent with the GWAS of lipid levels on the basis of which we constructed genetic instrumental variables[17], rank-based inverse normal transformed data were used in all analyses.

## Cell counts
White blood cell counts (WBC), i.e. neutrophils, lymphocytes, monocytes, eosinophils, and basophils, were measured by the standard WBC differential as part of the complete blood count (CBC). However, a minority of samples were lacking CBC measurements (15%) or did not differentiate between granulocyte subtypes (neutrophils, eosinophils, and basophils; 37%). Therefore, WBC and red blood cell counts were imputed for these samples from the RNA-seq data using the workflow found at: https://molepi.github.io/DNAmArray_workflow/05_Predict.html. The correlation between predicted and measured cell types was 0.91 for lymphocytes, 0.89 for neutrophils, 0.73 for monocytes, and 0.73 for eosinophils.

## Genotypes
Genotypes were measured individually per cohort (for data generation details see Tigchelaar et al.[38] for LL, Deelen et al. for LLS[44], Willemsen et al. for NTR[40], and Hofman et al. for RS[45]), but imputed centrally. In brief, the genotypes were harmonized (Genotype Harmonizer[46]), and imputed (Impute2[47]) using GoNL5[48] as a reference. Genotypes with an imputation info-score <0.5, Hardy–Weinberg equilibrium $P$-value < $10^{-4}$, call rate <95%, or minor allele frequency <0.05 were removed. In total, 5.2 million genotypes were available in all cohorts.

## Transcription
Total RNA libraries were generated using the TruSeq v2 library protocol and 2 × 50-bp paired-end sequencing was performed on the Illumina Hiseq2000. Reads passing Illumina's Chastity filter were produced using CASAVA and quality control was done with FastQC v0.10.1 (http://www.bioinformatics.babraham.ac.uk/projects/fastqc/), cutadapt v1.1 (adapter trimming[49]), and Sickle v1.2 (removal of low-quality read ends, https://github.com/najoshi/sickle). Reads were aligned to the human genome (build NCBI37, https://www.ncbi.nlm.nih.gov/assembly/GCF_000001405.13/) using STAR v2.3.0e[50]. Gene quantifications were obtained as the total number of reads that aligned to the exons of a gene as annotated by Ensembl v.71 (https://www.ensembl.info/2013/04/11/ensembl-71-has-been-released/). Subsequently, genes with zero reads in at least 20% of the samples were removed, gene counts were normalized with the TMM method using edgeR v3.28.1[51] and a rank-based inverse normal transformation was used to counteract deviations from normality and limit outliers. The final data set consists of 17,740 genes.

## Statistics
All analyses were performed in R v4.0.3, except for the clusterProfiler analysis, which was done in R v4.2.0.

For each cohort, we performed a transcriptome-wide analysis on lipid levels using cate v1.1.1 (https://cran.r-project.org/web/packages/cate/index.html) to estimate hidden confounders independent of the

other variables in the model. Test statistics were corrected for bias and inflation using bacon v1.14.0[52]. The following linear regression model was used for each gene:

$$\begin{aligned} \text{Transcription} = {}&\beta_0 + \beta_1{}^*\text{lipid level} + \beta_2{}^*\text{age} + \beta_3{}^*\text{sex} + \beta_4{}^*\text{WBC} \\ &+ \beta_5{}^*\text{RBC} + \beta_6{}^*\%\text{monocytes} + \beta_7{}^*\%\text{lymphocytes} + \beta_8 \\ &{}^*\%\text{neutrophils} + \beta_9{}^*\%\text{eosinophils} + \beta_{10}{}^*\text{hidden}_1 + \beta_{11} \\ &{}^*\text{hidden}_2 + \beta_{12}{}^*\text{hidden}_3 + \beta_{13}{}^*\text{hidden}_4 + \beta_{14}{}^*\text{hidden}_5 + \varepsilon \end{aligned} \quad (1)$$

Since the sum of all cell type fractions was 100%, one of the cell types (basophils) was not included as a covariate to prevent collinearity; the effect of the excluded cell type is captured in the intercept and thus corrected for implicitly. Nota bene, omitting eosinophils instead of basophils did not meaningfully alter the results (Supplementary Fig. 5). The results of each cohort were combined using a fixed-effect meta-analysis, and P-values were adjusted for 17,740 tests, i.e. the number of genes, using the Bonferroni method. Sensitivity analyses were performed with the potential confounder, i.e. either smoking behavior or lipid-lowering medication use, as extra covariate in the model. Smoking behavior and lipid-lowering medication traits were based on questionnaire data. Smoking behavior ($N = 3058$) was defined as never smoker, former smoker, and current smoker, and lipid-lowering medication use ($N = 3004$) was defined as user and non-user.

GIVs were created for TG, HDL-C, and LDL-C using genetic variants reported in GWAS that were >1 Mb apart and nearly independent ($r^2 < 0.10$)[17]. LL, NTR, and RS were part of this effort, with a maximum potential overlap of 2235 of 188,577 individuals. The following equation was used:

$$\text{GI} = \beta_1{}^*\text{dosage}_1 + \beta_2{}^*\text{dosage}_2 + \cdots + \beta_n{}^*\text{dosage}_n \quad (2)$$

In this equation $\beta$ is the GWAS regression estimate. The GIVs were scaled to mean 0 and standard deviation 1.

For each cohort, MR was performed to estimate the effect of lipid levels on transcription using the Wald method[53] with the GIVs as proxies for lipid levels. The association between GIV and transcription, which is required for this method, was calculated using a transcriptome-wide analysis as described previously, but with the following model for each gene:

$$\begin{aligned} \text{Transcription} = {}&\beta_0 + \beta_1{}^*\text{GI} + \beta_2{}^*\text{age} + \beta_3{}^*\text{sex} + \beta_4{}^*\text{WBC} + \beta_5{}^*\text{RBC} + \beta_6 \\ &{}^*\%\text{monocytes} + \beta_7{}^*\%\text{lymphocytes} + \beta_8{}^*\%\text{neutrophils} + \beta_9 \\ &{}^*\%\text{eosinophils} + \beta_{10}{}^*\text{hidden}_1 + \beta_{11}{}^*\text{hidden}_2 + \beta_{12}{}^*\text{hidden}_3 \\ &+ \beta_{13}{}^*\text{hidden}_4 + \beta_{14}{}^*\text{hidden}_5 + \varepsilon \end{aligned} \quad (3)$$

Robust standard errors were calculated using Fieller's theorem[54] and P-values were adjusted for multiple testing, i.e. the number of TWAS associations per lipid, using the Benjamini–Hochberg method at 5% FDR.

Pleiotropic genetic variants were detected using a Cochran's Q-test, which was adapted for use in MR[15], and iteratively removed from the GIVs until no evidence of pleiotropy remained ($P_Q > 0.5$). The association between genetic variation and transcription for each variant in the GIV, which is required for this method and for the method based on Egger regression used to identify residual directional pleiotropy[19], was calculated using a transcriptome-wide analysis as described previously, but with the following linear regression model for each gene:

$$\begin{aligned} \text{Transcription} = {}&\beta_0 + \beta_1{}^*\text{variant} + \beta_2{}^*\text{age} + \beta_3{}^*\text{sex} + \beta_4{}^*\text{WBC} + \beta_5{}^*\text{RBC} \\ &+ \beta_6{}^*\%\text{monocytes} + \beta_7{}^*\%\text{lymphocytes} + \beta_8{}^*\%\text{neutrophils} \\ &+ \beta_9{}^*\%\text{eosinophils} + \varepsilon \end{aligned} \quad (4)$$

Sensitivity analyses were performed using multivariable MR[20] with either (1) the dosage of the cis-expression QTLs in the GIVs as an extra covariate in the MR model to adjust for direct pleiotropy, or (2) the potential pleiotropic GIV as an extra covariate in the MR model to adjust for pleiotropy through a parallel path. GIVs for BMI, systolic blood pressure, and diastolic blood pressure were constructed using public GWAS data[21,22] with the same procedure as for the lipid GIVs.

Power calculations[55] were performed using code obtained from https://github.com/kn3in/mRnd.

To evaluate the opposite direction of effect, i.e. whether transcription affects lipid levels, we used for each gene the strongest associating cis-expression QTL[18] as a proxy for gene expression and used the same Wald method-based MR approach as described previously. All BIOS cohorts were part of the expression QTL GWAS, with a maximum potential overlap of 3229 of 31,684 participants.

Pathway enrichment using a one-sided Fisher's exact test was performed using clusterProfiler v4.4.0[56] with a background set of 17,740 genes that were expressed in our data. The 10 human pathway databases BioPlanet 2019, WikiPathways 2019 Human, KEGG 2019 Human, Elsevier Pathway Collection, BioCarta 2015, Reactome 2016, HumanCyc 2016, NCI-Nature 2016, Panther 2016 and MSigDB Hallmark 2020 were downloaded from https://maayanlab.cloud/Enrichr/#libraries and queried using gene symbols, with 39 of 55 queried genes present in at least 1 database. Multiple testing using the Benjamini–Hochberg method at 5% FDR was performed over the combined results from the 10 databases.

Finally, we assessed the effects of genes affected by TG on allergy phenotypes and the incidence of several chronic diseases using two-sample MR[57]. We selected several large allergy GWAS studies for their precise estimates, i.e. a combined allergic disease phenotype (asthma and/or hay fever and/or eczema)[26], incidence of childhood-onset eczema[25] and incidence of adult-onset eczema[25], and a smaller but more specific GWAS of IgE levels[27]. We also selected several large chronic disease incidence GWAS studies for their link with lipids and/or inflammation, namely coronary artery disease[28], myocardial infarction[28], and rheumatoid arthritis[29]. We used for each gene the strongest associating cis-expression QTL[18] as a proxy for gene expression and used the same Wald method-based MR approach as described previously.

### Reporting summary
Further information on research design is available in the Nature Portfolio Reporting Summary linked to this article.

## Data availability
The RNA sequencing data generated in this study and phenotypes age, sex, and cell types have been deposited in the EGA database under accession code EGAS00001001077, and data access procedures are available at https://www.bbmri.nl/acquisition-use-analyze/bios. The genotype and other phenotypes, data are governed by the respective biobanks. Access can be requested according to the procedures established by the biobanks, with restrictions imposed by the respective institutional review boards and Dutch law. The Supplementary figures generated in this study are provided in the Supplementary Information file and the Supplementary Tables are provided as separate Supplementary Data files. Public databases used in this study include the human genome (build NCBI37, https://www.ncbi.nlm.nih.gov/assembly/GCF_000001405.13/), Ensembl v.71 (https://www.ensembl.info/2013/04/11/ensembl-71-has-been-released/), and pathway databases BioPlanet 2019, WikiPathways 2019 Human, KEGG 2019 Human, Elsevier Pathway Collection, BioCarta 2015, Reactome 2016, HumanCyc 2016, NCI-Nature 2016, Panther 2016 and MSigDB Hallmark 2020 downloaded from https://maayanlab.cloud/Enrichr/#libraries. Public datasets used in this study include those from Willer et al.[17], Võsa et al.[18], Locke et al.[21], Warren et al.[22], Monaco et al.[24], Ferreira et al.[25,26], Nikpay et al.[28] and Okada et al.[29].

## Code availability

R-code used to preprocess the data is available at https://github.com/bbmri-nl/BBMRIomics and R-code used to perform the analyses is available at www.github.com/kfdekkers/twasmr[58].

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

## Acknowledgements

This research was financially supported by BBMRI-NL, a Research Infrastructure financed by the Dutch government (NWO, numbers 184.021.007 and 184.033.111). We acknowledge the support from the Netherlands CardioVascular Research Initiative (the Dutch Heart Foundation, Dutch Federation of University Medical Centres, the Netherlands Organisation for Health Research and Development, and the Royal Netherlands Academy of Sciences) for the GENIUS project Generating the best evidence-based pharmaceutical targets for atherosclerosis (CVON2011-19, CVON2017-20). This work was carried out on the Dutch national e-infrastructure with the support of SURF Cooperative.

## Author contributions

K.F.D., J.W.J., and B.T.H. designed the study. B.I.O.S., M.v.I., M.A.I., M.v.G., J.H.V., L.F., D.I.B., P.E.S., B.T.H. contributed to establishing the data infrastructure. K.F.D., R.C.S., and B.T.H. performed the analyses. K.F.D. and B.T.H. drafted the manuscript. R.C.S., A.I.F., M.v.I., M.A.I., M.M.J.v.G., J.H.V., L.F., D.I.B., P.E.S., and J.W.J. were involved in critical revisions of the manuscript. All authors read and approved the final manuscript.

## Competing interests

The authors declare no competing interests.

## Additional information

## BIOS consortium

Maarten van Iterson[1], Marleen M. J. van Greevenbroek [6], Jan H. Veldink [7], Lude Franke [8], Dorret I. Boomsma [9], P. Eline Slagboom [1] & Bastiaan T. Heijmans [1]✉

A full list of members and their affiliations appears in the Supplementary Information.

