## [Peer Review File · Nature Communications]

Lipid-induced transcriptomic changes in blood link to lipid metabolism and allergic responseREVIEWER COMMENTS

Reviewer #1 (Remarks to the Author):

Lipid-induced transcriptomic changes in blood are involved in lipid metabolism and allergic response: a Mendelian randomization study
Dekkers et al.

Reviewer Comments

The Authors aimed to evaluate the causality between lipid levels (TG, HDL, LDL) and blood transcription and its direction (with focus on the effect of lipid levels on transcription). Using data of the BIOS consortium, the Authors thus conducted a transcriptome-wide association study of lipid levels and Mendelian Randomization (MR) analysis. Beside extensive sensitivity analyses, the Authors conducted downstream analyses including enrichment analysis and 2-sample MR analysis with respect to specific allergy phenotypes.

The Authors did a thorough job to address the quite complex research aim. However, the description of methods and results is sometimes quite crude and cryptic. I think only for people with direct insight into the project, the presentation may be fully understandable. So, to avoid guessing and misinterpretation on the Readers' side more explanations and details are required. In the following I will provide comments in order of analysis workflow which bases on my interpretation (or guessing):

1 Studies and Data

Methods/Cohorts: Data used in the main analysis were obtained from cohort studies of the BIOS consortium. The Authors state samples used for the different measurements were obtained simultaneously from participants I assume. But how about the conduct of measurements? It seems that measurements were done within each study cohort separately. But maybe, the Authors could provide more details, especially whether measurements were done in different labs and/or at different time points?

Methods/Lipids: How skewed are the lipid measurements requiring such extensive transformation (inverse normal transformation of ranks)? The use of fasting and non-fasting samples may also be one reason requesting this kind of transformation. Were the correlation coefficients (Results: Lines 79-80) calculated based on transformed lipid measurements?

Methods/Cell counts: I am not familiar with imputation of cell counts and its quality. What is the imputation quality?

Methods/Genotypes: The Authors may provide a statement on number of variants used in the analysis.

2 Transcriptome-wide analysis of lipid levels

Methods/Statistics (Line 351): Did the Authors check heterogeneity when meta-analysing data or did they just assumed homogeneity allowing for the use of a fixed-effect model?

Methods/Statistics (Lines 351-352): To correct for multiple testing a Bonferroni correction was applied. Since it is not further described, please add what exactly was corrected for (number of genes? number of lipid traits?) and the exact factor with whom p-values were adjusted.

Table S1: Please provide criterion for the selection of presented results in the legend.

Figure 1A: Please explain horizontal reference line.

Figure 1B: I think a better representation of this data can be achieved with a Venn diagram. Similarly, for Figure 2B.

Methods/Statistics (Lines 352-353) and Figure S1: Description of this sensitivity analysis is quite short. Were smoking behavior and lipid-lowering medication use the only two variables checked in this part? How are they defined? Were data available in all studies? Please add the meaning of grey points in Figure S1 to the legend.

3 Construction of GIV for lipids

Methods/Statistics (Lines 354-357) and Results (Lines 97-99): For the construction of GIVs for lipids, the Authors took advantage of publicly available data. Please provide general information on this study (incl. trait definition). Is there an overlap with the BIOS consortium? How were the single variants selected? What does esx in the formula to calculate GIV mean? I would assume the effect estimate from that published GWAS. Please extend Table S2 by general information (chr, position) and association statistics reported by the published GWAS. Please also describe at this point the overlap in selected variants for the different lipid traits.

Results (Lines 101-105): Obtained GIVs were assessed for association with potential confounders (Table S4). While the text suggests that p-values are presented, the legend of Table S4 suggests differently. Please revise and clarify.

4 MR analysis

Methods/Statistics (Lines 366-367): For the conduct of the MR analysis, the conduct of a transcriptome-wide analysis of transcription using GIVs of lipids was requested. To ascertain statistical significance, the Benjamini-Hochberg (BH) procedure was used. This procedure belongs to the group of methods addressing false discovery rate (FDR) why the Authors in Results likely refer to pFDR (e.g., Line 112). Neither the abbreviation for FDR is introduced nor the connection to the BH method is provided. Please adjust at all instances throughout the manuscript.

Results (Lines 112-113): Did the exclusion of variants affect previous statements on strength and association with confounders? Please add selection criterion to the legend of Table S5.

Results (Lines 116-118) and Methods (Lines 378-379): As discussed by many authors/statisticians (e.g. Dziak et al 2020, PMID: 32523323), post-hoc power calculations as presented in this paper are not valid. This part as well as the respective part in Discussion (Lines 253-255) need to be deleted and the topic differently approached.

Results (Lines 121-125): This 2nd part of the MR analysis refers to the MR analysis of the opposite direction and focused on the findings of the 1st part of MR analysis. Together with the previous MR part, they present the bidirectional analysis.

Please revise title of Table S6 presenting results regarding the opposite (or reverse) direction.

Furthermore, please add – similarly as for the lipid GIVs – information on the published data from which instruments were selected by describing that study, providing information on instruments in another supplementary table and describing potential overlap with BIOS consortium.

Furthermore, it is unclear why 64 genes were assessed as the 1st part of the MR analysis reported 62 genes? Why are some rows in Table S6 empty? In addition, I would suggest to report all assessed relations or alternatively add the selection criterion to the legend.

5 Sensitivity analysis regarding pleiotropy

Results (Lines 133-140): This part is too cryptic. To allow better understanding please make sure that Readers can find the respective part in the Methods section by, for example, mention in Lines 371-376 the term "direct adjustment" or alternatively, the phrase "assessment of residual pleiotropy" in the Results part, or both in order to give some cross-reference.

Please expand the Methods part (Lines 371-376) as I do not understand what the Authors have actually done, also with respect to Figure S3. From the presented formula I would assume that per transcription, several models were fitted (for each single variant) but this does not fit to the display in Figure S3. From that figure I can also not deduce the jump to rs174546 mentioned in Results (Line 139).

Please also revise the legend in Figure S3 as the terms 'none', etc cannot be understood and explain what selection of results is actually displayed.

In general, I would suggest to add all findings (incl. those from other sensitivity analyses) to Table S5 if not done already.

Results (Lines 145-150): There is no explanation given how GIVs for BMI, systolic and diastolic blood pressure were defined. Please add with everything that needs to be presented.

Results (Lines 151-155): This whole chapter ends with the statement that 2 genes are excluded from further consideration. Please make this clearer. I assume you mean the two genes LYBD6B and CCNA1 marked in Table S5. This might be achieved by also including consequences such as this to Table S5.

6 Enrichment analysis

Methods (Lines 383-387): The Authors used Enrichr to assess pathway enrichment using 10 different databases. Since presented terms in Table S8 seem overlapping, the question arises why using all those 10 databases? How do they differ? Was the mapping done via gene name or Ensembl ID? Could everything mapped?

Results (Lines 163-189): Presentation of results are difficult to retrace only by having Table S8. I would thus suggest to expand Table S8 relating genes to lipid traits and terms (in text named as processes) to classifications (e.g., lipid metabolism). Any further support to deduce numbers stated in Results (e.g., 22 processes related to allergy, Line 175) can only be appreciated because of similarities of terms.

Figure 3B: If Table S8 is expanded it might also help to explain the grouping of genes not explained otherwise in the legend of this figure.

7 2-sample MR analysis related to allergy phenotypes

Methods (Lines 388-389): Please expand. Essentially everything is missing. Why those 4 phenotypes? What are these studies? How are traits defined? Selection of GIVs? ...

8 Discussion

Overall, the Discussion is thorough. The Authors stated under limitations that relevant disease outcomes were not available. What disease outcomes do the Authors miss? Is this meant in reference to allergy phenotypes?

Reviewer #2 (Remarks to the Author):

The authors present a novel approach to test for causality between biomarkers and transcription, and transcription and disease end points. The paper correlates standard lipid measurements with whole

blood transcriptome of 3229 individuals and uses Mendelian randomization approaches to identify 55 genes where triglyceride levels potentially cause transcriptional changes, some with links to allergic response.

The paper is mostly clear and has potentially interesting results. However, there are several points in analyses and interpretations of the results that would need clarification:

1. One of the cohorts LLS did not fast before lipid measurements. How different are the TG measures in LLS compared to other cohorts? It would be good to do sensitivity analysis of without LLS to make sure that the results are not driven by differences in fasting.
2. There seemed to be very limited information on how the GIVs were constructed. How were the variants chosen for GIVs? How did you account for the LD between variants?
3. The GIVs were tested against the potential confounders listed in Table S4. How were these confounders chosen? Lipid variants are known to associate with many other risk factors like BMI, WHR, blood pressure etc. and disease risks like CAD/stroke/T2D as well which may potentially also affect expression levels. These variables are also available in most if not all of the participating cohorts as well. These could also be tested in the cohorts or at least their effect could be discussed.
4. The two sample MR analysis on expression levels and allergy could be benchmarked against similar analysis of TG-associated genes and CAD & T2D. As allergies also typically has onset in early life, it is also not quite clear how existing allergies may affect expression levels of the genes measured later in life as in these cohorts. This could be discussed as well.
5. How was the considerable multiple testing burden controlled for in different stages of the analyses?

Minor comments:

- In Figure 1 Panel B, please fix the X-axis.
- In Discussion, line 230 please change "mutations" -> "variants"
- Discussion, line 249, please add "potential" to "This study the first study with sufficient statistical power to find potential causal relationships..."
- In Discussion, line 271, please remove the work "rigorous"

Reviewer #3 (Remarks to the Author):

In this study, Dekkers et al study the whole blood transcriptome of >3000 human subjects and use a mendelian randomization approach (more specifically GIV) to study which genes are transcribed under the influence of blood lipids through genetics. This is an elegant approach, allowing you to rule out potential environmental upregulation of lipid levels, and more importantly, study directionality. By doing this, they found 55 genes in immune cells to be genetically associated with triglycerides, which are genes in lipid metabolism pathways and allergic disease. The finding of genes in allergic disease are new and surprising, although the methods used (and more specifically the covariates added in the model) do rise some questions to the validity of these findings. Although the scope of my research is outside the use of MR methodology, I do think there are other important questions to be raised:

1. Why were 10 human pathway databases used to analyse the gene set? In fact, more pathways were found than genes. This feels quite like the fishing expedition. Did the authors correct for the multiple analyses that were performed?
2. The most striking and new result is the link between TG genotypes and genes in allergic disease. (It is quite obvious that genes in lipid metabolism are affected by TG) However, I am concerned that this is in fact an artifact of the covariates applied to (and/or left out of) the model. In the methods section, the authors describe that basophils were left out of the covariates that were corrected for due to the sum of 100%. This makes sense, but for sensitivity analysis (and the potential bias of the data), the same analysis should be repeated with eosinophils left out instead of basophils. Another option would be to work with numbers of cells and not % (those can be calculated using the WBC and the % together). Could the authors repeat the analysis with the raw numbers of cells and not the %? Are the genes

related to basophils then still upregulated?

3. Was sensitivity analysis performed on the origin of the data? Some cohorts described are general public cohorts, whereas other cohorts (especially the CVD/Diabetes cohort) are more specific. Did some cohorts skew the results? The same question arises for the fasting / non-fasting analysis of lipids

4. The authors start the abstract and introduction with an interest in priming of immune cells by lipids and trained immunity. However, by using an MR approach, the authors actually rule out the effects of lipids and environmental / temporary increase by using the genetic approach. All immune cells will also intrinsically be affected by these genetic variations which could lead to the changes in gene expression rather than due to the rise in TG levels in the circulation. Furthermore, the authors do not discuss this phenomenon of trained immunity or priming at all anymore in the rest of the paper. Do the authors believe they have added to the understanding of trained immunity by lipids? If not, it should be removed from the abstract and introduction as it is misleading.

5. Following up on the fourth point: the 55 genes that were found to be influenced by genetics are only a small part of the 496 initially associated TG genes. Does this mean that the other genes are more environmentally associated? That could be an interesting point of view, as environmentally associated genes are potentially more reversible than genetics. Did the authors look at the difference between the two findings and what pathways are associated with those genes?

Minor comments

Line 172-175: The authors mention three genes, after which two genes are actually explained. What is the third additional gene?

Line 253: This study the first study – “is” is missing

Figure 1b: the x-axis font size is too big: words are overlapping

Reviewer #1 (Remarks to the Author):

Lipid-induced transcriptomic changes in blood are involved in lipid metabolism and allergic response: a Mendelian randomization study

Dekkers et al.

Reviewer Comments

The Authors aimed to evaluate the causality between lipid levels (TG, HDL, LDL) and blood transcription and its direction (with focus on the effect of lipid levels on transcription). Using data of the BIOS consortium, the Authors thus conducted a transcriptome-wide association study of lipid levels and Mendelian Randomization (MR) analysis. Beside extensive sensitivity analyses, the Authors conducted downstream analyses including enrichment analysis and 2-sample MR analysis with respect to specific allergy phenotypes.

The Authors did a thorough job to address the quite complex research aim. However, the description of methods and results is sometimes quite crude and cryptic. I think only for people with direct insight into the project, the presentation may be fully understandable. So, to avoid guessing and misinterpretation on the Readers' side more explanations and details are required. In the following I will provide comments in order of analysis workflow which bases on my interpretation (or guessing):

We thank the Reviewer for the valuable comments and, upon re-reading, agree that the Results and Method sections could be difficult to follow for researchers not intimately familiar with the Mendelian randomization methodology. We have adjusted the sections based on the Reviewer's concerns.

1 Studies and Data

Methods/Cohorts: Data used in the main analysis were obtained from cohort studies of the BIOS consortium. The Authors state samples used for the different measurements were obtained simultaneously from participants I assume. But how about the conduct of measurements? It seems that measurements were done within each study cohort separately. But maybe, the Authors could provide more details, especially whether measurements were done in different labs and/or at different time points?

We added this information to the manuscript. RNA sequencing was done at the same core facility for all cohorts. Also RNA-seq pre-processing and imputation was done centrally. However, genotyping arrays, lipid and cell counts measurements, were performed individually by the cohorts prior to the start of the BIOS Consortium.

Added to Methods/Cohorts:

The measurements of the samples for the genotyping arrays, lipids and cell counts data were performed individually by the cohorts. All RNA-seq data were generated centrally

within the BIOS Consortium by the Human Genotyping facility (HugeF) of ErasmusMC, the Netherlands (<http://www.glimdna.org/>).

Added to Methods/Genotypes:

Genotypes were measured **individually** per cohort (for data generation details see Tigchelaar et al. (39) for LL , Deelen et al. for LLS (45), Willemsen et al. for NTR (41), and Hofman et al. for RS (46)), **but imputed centrally**.

Methods/Lipids: How skewed are the lipid measurements requiring such extensive transformation (inverse normal transformation of ranks)? The use of fasting and non-fasting samples may also be one reason requesting this kind of transformation. Were the correlation coefficients (Results: Lines 79-80) calculated based on transformed lipid measurements?

In our data, LDL-C and HDL-C levels were approximately normally distributed, while TG level was positively skewed as expected (see Figure below with distributions of non-transformed data). The GWAS of lipids levels from which we took the effect sizes for our genetic instruments used rank-based inverse normal transformation to address non-normality [1]. For consistency, we implemented the same approach. As to the potential effect of fasting on our results, we did not observe any obvious difference in the distribution of lipids between non-fasting (LLS) and fasting (other biobanks) cohorts (see Reviewer Figure 1 below). Importantly, the results obtained from the non-fasting cohort were in line with those from the other cohorts as is apparent from the fact that for 906

associations observed in the transcriptome-wide analysis, the direction of effects were highly consistent between cohorts: for 84% (757/906) of genes the direction was the same for all 6 cohorts and for 100% (905/906) of genes the non-fasting cohort showed the same direction as the combined estimate of all cohorts (Table S1).

Reviewer Figure 1. Distributions of non-transformed lipid levels for the six cohorts of the BIOS study.

The correlation coefficients were based on the transformed data. The correlations for the non-transformed data were: TG,HDL-C: -0.43, TG,LDL-C: 0.34, HDL-C,LDL-C: -0.13.

Added to the Results:

Evidence for the associations was generally consistent across cohorts: for 757 of 906 associations the direction was consistent for all 6 cohorts, for 143 associations the

direction was consistent for 5 of 6 cohorts, for 5 associations the direction was consistent for 4 of 6 cohorts, and for 1 association the direction was consistent for 3 of 6 cohorts (Table S1).

Added to the Methods:

To address non-normality of the distributions of lipid levels and to be consistent with the GWAS of lipid levels on the basis of which we constructed genetic instrumental variables (17), rank-based inverse normal transformed data were used in all analyses.

Methods/Cell counts: I am not familiar with imputation of cell counts and its quality. What is the imputation quality?

Imputation of cell counts is common in studies that have gene expression or epigenetic data measured in blood samples, but no cell count measurements, since the proportion of blood cell types differs in each person and different blood cell types have different gene expression and epigenetic profiles. A workflow for the cell imputation methodology can be found at https://molepi.github.io/DNAArray_workflow/05_Predict.html. The resulting quality of the imputation was good, i.e. the correlation between predicted and measured cell types was 0.91 for lymphocytes, 0.89 for neutrophils, 0.73 for monocytes, and 0.73 for eosinophils. Added to the methods:

The correlation between predicted and measured cell types was 0.91 for lymphocytes, 0.89 for neutrophils, 0.73 for monocytes, and 0.73 for eosinophils.

Methods/Genotypes: The Authors may provide a statement on number of variants used in the analysis.

We added the number of genetic variants available for analysis to the Methods section.

Added to the Methods:

In total 5.2 million genotypes were available.

2 Transcriptome-wide analysis of lipid levels

Methods/Statistics (Line 351): Did the Authors check heterogeneity when meta-analysing data or did they just assumed homogeneity allowing for the use of a fixed-effect model?

To provide insight into the level of heterogeneity, we added information about the direction of association of the cohorts to Table S1. We observed that the cohorts have a consistent direction of association for 757 of 906 associations, for 143 associations 5 cohorts have a consistent direction of association (the small biobanks PAN and CODAM are most often the odd ones out), for 5 associations 4 cohorts have a consistent direction of association (CODAM and PAN are the odd ones out) and for 1 association 3 cohorts have a consistent direction of association (CODAM, PAN and LLS are negatively associated and LL, NTR and RS are positively associated). We therefore believe that heterogeneity played at most a minor role in the results.

Added to the Results:

Evidence for the associations was generally consistent across cohorts: for 757 of 906 associations the direction was consistent for all 6 cohorts, for 143 associations the direction was consistent for 5 of 6 cohorts, for 5 associations the direction was consistent for 4 of 6 cohorts, and for 1 association the direction was consistent for 3 of 6 cohorts (Table S1).

Added to Table S1 the column Direction containing the direction of associations for the cohorts.

Added to the legend of Table S1:

Column Direction contains the direction of association for cohorts CODAM, LL, LLS, NTR and PAN, respectively.

Methods/Statistics (Lines 351-352): To correct for multiple testing a Bonferroni correction was applied. Since it is not further described, please add what exactly was corrected for (number of genes? number of lipid traits?) and the exact factor with whom p-values were adjusted.

The P-values were corrected for the number of genes, which we now make clear in the text.

Added to the Methods:

The results of each cohort were combined using a fixed-effect meta-analysis and P-values were adjusted for 17,740 tests, i.e. the number of genes, using Bonferroni's method.

Table S1: Please provide criterion for the selection of presented results in the legend.

We show only the genome-wide significant results based on the Bonferroni correction.

Added to the legend of Table S1:

Presented are genome-wide significant results after adjusting for 17,740 tests, i.e. the number of genes, using the Bonferroni method.

Figure 1A: Please explain horizontal reference line.

The horizontal line represents the Bonferroni threshold.

Added to the legend of Figure 1A:

Dashed horizontal line represents the Bonferroni threshold based on 17,740 tests, i.e. the number of genes.

Figure 1B: I think a better representation of this data can be achieved with a Venn diagram. Similarly, for Figure 2B.

Actually, we originally tried to represent the results using a Venn Diagram but felt it did not work well. We tried both conventional Venn diagrams and area proportional Venn diagrams, but irrespectively the Venn diagram did not adequately represent the data because of the larger number of categories with large differences in the number of genes (much lower for LDL-C than HDL-C and TG).

Methods/Statistics (Lines 352-353) and Figure S1: Description of this sensitivity analysis is quite short. Were smoking behavior and lipid-lowering medication use the only two variables checked in this part? How are they defined? Were data available in all studies? Please add the meaning of grey points in Figure S1 to the legend.

Our strategy for confounders and mediators was as follows:

We adjusted our models for common gene expression covariates age, sex and cell counts to get more precise estimates. We also verified that covariates had no association with the GIVs to make sure they could be used in the MR models.

We further assessed the effect of smoking behavior and lipid-lowering medication on our TWAS results in a sensitivity analysis, since this data was not available for each participant (smoking behavior N = 3058 of 3229; lipid-lowering medication N = 3004 of

3229, missing for PAN). The sensitivity analysis showed that these factors did not influence our results (Figure S1).

Finally, we assessed the influence of blood pressure and BMI in the MR analysis. We were hesitant to include these variables in the TWAS analysis, since they are potential mediators. However, we wanted to exclude their potential pleiotropic effect in the MR analyses.

Together, we feel that this represents an adequate assessment of potential confounders and mediators.

Results

Similarly, we constructed GIVs for other potential pleiotropic factors, namely systolic blood pressure, diastolic blood pressure, and BMI, using public GWAS data (21,22) and subsequently applied the same procedure as implemented for the lipid GIVs (Figure S4). This analysis did not detect any further pleiotropic associations.

Added to the Methods:

Sensitivity analyses were performed with the potential confounder, *i.e. either smoking behavior or lipid-lowering medication use*, as extra covariate in the model. *Smoking behavior and lipid-lowering medication traits were based on questionnaire data.*

Smoking behavior (N = 3058) was defined as never smoker, former smoker and current smoker, and lipid-lowering medication use (N = 3004) was defined as user and non-user.

Added to Figure S1:

Points depicted in color represent genome-wide significant associations in the model without adjustment for smoking behavior or lipid-lowering medication use.

3 Construction of GIV for lipids

Methods/Statistics (Lines 354-357) and Results (Lines 97-99): For the construction of GIVs for lipids, the Authors took advantage of publicly available data. Please provide general information on this study (incl. trait definition). Is there an overlap with the BIOS consortium? How were the single variants selected? What does esx in the formula to calculate GIV mean? I would assume the effect estimate from that published GWAS. Please extend Table S2 by general information (chr, position) and association statistics reported by the published GWAS. Please also describe at this point the overlap in selected variants for the different lipid traits.

We constructed the GIVs using independent genetic variants associated with lipid levels based on a large GWAS on lipids [1]. LL, NTR and RS were part of this effort, with a maximum potential overlap of 1.2% (2,235 of a total of 188,577 samples included in the GWAS). We think this represents a sufficiently small fraction that has no meaningful effect on our results. Variants reported in the GWAS were >1 Mb apart and nearly independent ($r^2 < 0.10$). As the Reviewer guessed, esx referred to the mean GWAS

regression estimate. Indeed, the notation is not clear and we changed it to βx in line with other equations and explained its meaning.

Added to the Results:

To infer causal relationships using MR, we first constructed weighted genetic instrumental variables (GIVs) for blood lipids from genetic variants reported in a genome-wide association study of lipid levels among 188,577 individuals (17) (TG: 40 variants, HDL-C: 69 variants, LDL-C: 57 variants; all variants were available in the current study; Table S2). The GIVs were strongly associated with their respective lipid levels in our own study (F-stat > 134, P-value < 10⁻³¹; Table S3), although they explained a minor proportion of the total variance (R² = 4.0% – 6.4%). There was overlap between the variants of the lipid GIVs. Of the variants, 18/40, 49/69 and 44/57 were unique for TG, HDL-C and LDL-C, respectively.

Added to the Results:

Finally, effect sizes were generally not sensitive to further adjustment for GIVs of the other lipids using a multivariable MR analysis (20). This is important because there was overlap between the lipid GIVs and was achieved by adjusting the effect of the lipid of interest for the GIVs of the other lipids.

Added to the Methods:

GIVs were created for TG, HDL-C and LDL-C using genetic variants reported in a GWAS among 188,577 individuals that were >1 Mb apart and nearly independent ($r^2 <$

0.10) (17). LL, NTR and RS were part of this effort, with a maximum potential overlap of 2235 of 188,577 individuals. The following equation was used:

(See manuscript for equation where βx is the GWAS regression estimate.)

Added to Table S2:

Chromosome, position and GWAS association statistics.

Added to the legend of Table S2:

Table S2. Genetic variants used to create genetic instruments. Variants with P-value $< 5 \times 10^{-8}$ that were >1 Mb apart and nearly independent ($r^2 < 0.10$) reported in a GWAS on lipids (17).

Results (Lines 101-105): Obtained GIVs were assessed for association with potential confounders (Table S4). While the text suggests that p-values are presented, the legend of Table S4 suggests differently. Please revise and clarify.

We thank the Reviewer for pointing out the error in the table legend. Presented in Table S4 are the P-values of the associations between GIV and potential confounder.

Changed the legend of Table S4 to:

Presented are the P-values of the linear regression model.

4 MR analysis

Methods/Statistics (Lines 366-367): For the conduct of the MR analysis, the conduct of a transcriptome-wide analysis of transcription using GIVs of lipids was requested. To ascertain statistical significance, the Benjamini-Hochberg (BH) procedure was used. This procedure belongs to the group of methods addressing false discovery rate (FDR) why the Authors in Results likely refer to pFDR (e.g., Line 112). Neither the abbreviation for FDR is introduced nor the connection to the BH method is provided. Please adjust at all instances throughout the manuscript.

We have changed each instance of “using the Benjamini-Hochberg method” to “using the Benjamini-Hochberg method at a 5% false discovery rate” or “using the Benjamini-Hochberg method at 5% FDR”.

An example in the Results:

We found evidence of an effect of TG on 56 genes, of HDL-C on 6 genes and of LDL-C on 0 genes **after adjustment for multiple testing using the Benjamini-Hochberg method at a 5% false discovery rate ($P_{FDR} < 0.05$, Table S5).**

Results (Lines 112-113): Did the exclusion of variants effect previous statements on strength and association with confounders? Please add selection criterion to the legend of Table S5.

As can be seen from Reviewer Tables 1 and 2, removing pleiotropic variants from the GIVs does not substantially alter the strength of the instrument or the association with confounders. Removing rs964184, rs7241918, rs3136441, rs11246602 and rs13107325 from the HDL-C GIV does result in a nominally significant association with eosinophil cell counts (P-value = 0.04), however this is weak when compared to the association of the HDL-C GIV with HDL-C levels (P-value = 5.6×10^{-31}).

Reviewer Table 1. Strength of lipid GIV with potential pleiotropic variants removed.

Lipid	Removed.rsID	Estimate	CI.Lower	CI.Upper	P.value	R.squared	F.statistic
TG	None	0.21	0.18	0.25	5.60E-31	0.040	134
TG	rs1121980	0.22	0.18	0.25	1.32E-31	0.041	137
TG	rs11776767,rs44 2177	0.21	0.18	0.25	2.12E-30	0.039	132
TG	rs1532085,rs174 546	0.21	0.17	0.25	1.47E-29	0.038	128
TG	rs1532085,rs213 1925	0.21	0.17	0.24	2.12E-29	0.038	126
TG	rs1532085,rs968 6661,rs2954029	0.21	0.17	0.24	1.30E-28	0.037	123
TG	rs174546,rs1532 085	0.21	0.17	0.25	1.47E-29	0.038	128

TG	rs3764261,rs116 49653	0.21	0.18	0.25	8.76E-31	0.040	133
TG	rs3764261,rs127 48152	0.21	0.18	0.25	4.37E-31	0.040	134
TG	rs3764261,rs241 2710	0.22	0.18	0.25	6.87E-32	0.041	139
TG	rs442177	0.21	0.18	0.25	2.47E-30	0.039	132
HDL-C	None	0.26	0.22	0.29	3.29E-50	0.064	220
HDL-C	rs174546	0.26	0.22	0.29	1.32E-50	0.064	222
HDL-C	rs1883025	0.25	0.22	0.29	2.44E-48	0.062	212
HDL-C	rs2954029	0.25	0.22	0.29	2.50E-49	0.063	216
HDL-C	rs499974	0.26	0.22	0.29	8.00E-50	0.063	218
HDL-C	rs964184,rs7241 918,rs3136441,r s11246602,rs13 107325	0.25	0.22	0.28	2.88E-47	0.061	208

Reviewer Table 2. Association P-values of lipid GIV with potential confounders.

Lipid	Removed.rsID	Age	Sex	Neut	Lymph	Mono	Eos	WBC	RBC
-------	--------------	-----	-----	------	-------	------	-----	-----	-----

TG	None	0.77	0.29	0.45	0.90	0.54	0.82	0.94	0.78
TG	rs1121980	0.80	0.28	0.49	0.97	0.52	0.80	0.95	0.80
TG	rs11776767,rs442 177	0.78	0.38	0.54	0.98	0.49	0.84	0.88	0.79
TG	rs1532085,rs1745 46	0.69	0.38	0.40	0.85	0.47	0.72	0.97	0.79
TG	rs1532085,rs2131 925	0.92	0.29	0.43	0.84	0.48	0.73	0.97	0.63
TG	rs1532085,rs9686 661,rs2954029	0.75	0.34	0.37	0.89	0.29	0.81	0.96	0.82
TG	rs174546,rs15320 85	0.69	0.38	0.40	0.85	0.47	0.72	0.97	0.79
TG	rs3764261,rs1164 9653	0.76	0.29	0.36	0.76	0.49	0.79	0.86	0.72
TG	rs3764261,rs1274 8152	0.67	0.34	0.53	0.99	0.50	0.69	0.84	0.83
TG	rs3764261,rs2412 710	0.89	0.20	0.27	0.64	0.40	0.63	0.99	0.68
TG	rs442177	0.79	0.36	0.49	0.96	0.52	0.81	0.95	0.78
HDL-C	None	0.76	0.63	0.58	0.82	0.17	0.11	0.42	0.68
HDL-C	rs174546	0.83	0.73	0.58	0.82	0.16	0.11	0.46	0.65

HDL-C	rs1883025	0.64	0.71	0.61	0.82	0.20	0.11	0.44	0.72
HDL-C	rs2954029	0.78	0.67	0.61	0.80	0.25	0.10	0.45	0.68
HDL-C	rs499974	0.86	0.59	0.58	0.83	0.18	0.10	0.41	0.69
HDL-C	rs964184,rs72419 18,rs3136441,rs1 1246602,rs13107 325	0.90	0.61	0.40	0.69	0.09	0.04	0.48	0.34

Added to the legend of Table S5:

Presented are significant results after adjusting for 496 (TG), 284 (HDL-C) and 26 (LDL-C) tests using the Benjamini-Hochberg method at 5% FDR.

Results (Lines 116-118) and Methods (Lines 378-379): As discussed by many authors/statisticians (e.g. Dziak et al 2020, PMID: 32523323), post-hoc power calculations as presented in this paper are not valid. This part as well as the respective part in Discussion (Lines 253-255) need to be deleted and the topic differently approached.

We agree with the Reviewer that post-hoc power calculations are not an accurate way to calculate statistical power, however it does provide a reasonable estimate of the order of magnitude of the sample sizes that MR studies require. This power calculation

was requested by other reviewers in two of our other MR papers [3], [4] and while we were initially hesitant to add it based on the well-documented flaws, we now view this as a meaningful addition for readers.

Added to the Discussion:

While post-hoc power calculations are not a valid method to estimate the statistical power of a study (34), it does provide a reasonable estimate of the order of magnitude of the sample sizes that MR studies require.

Results (Lines 121-125): This 2nd part of the MR analysis refers to the MR analysis of the opposite direction and focused on the findings of the 1st part of MR analysis. Together with the previous MR part, they present the bidirectional analysis.

Please revise the title of Table S6 presenting results regarding the opposite (or reverse) direction. Furthermore, please add – similarly as for the lipid GIVs – information on the published data from which instruments were selected by describing that study, providing information on instruments in another supplementary table and describing potential overlap with BIOS consortium.

We selected the genetic variant most strongly associated with expression of the lipid-associated genes based on data from the eQTLGen consortium [1]. All BIOS cohorts were part of this effort, with a maximum potential overlap of 3,229 of 31,684 participants.

Added to the Methods:

All BIOS cohorts were part of the expression QTL GWAS, with a maximum potential overlap of 3229 of 31,684 participants.

Changed the title of Table S6:

Table S6. Results of the reverse MR analysis: transcription does not affect blood lipid levels.

Furthermore, it is unclear why 64 genes were assessed as the 1st part of the MR analysis reported 62 genes? Why are some rows in Table S6 empty? In addition, I would suggest to report all assessed relations or alternatively add the selection criterion to the legend.

We thank the Reviewer for noticing the mistake and changed 64 to 62. The empty rows represent genes for which an eQTL was not found in the eQTLGen consortium data. Presented in Table S6 are results for all lipid associated genes after adjusting for 496 (TG), 284 (HDL-C) and 26 (LDL-C) tests using the Benjamini-Hochberg method at 5% FDR.

Changed the Results:

We also extended the MR analysis to evaluate the opposite direction of effect, i.e. whether transcription of the 62 genes observed in the MR analysis can affect lipid levels, a procedure known as bidirectional MR.

Added to the legend of Table S6:

Presented are results for the 56 genes affected by TG and the 6 genes affected by HDL-C in the forward MR analysis, see Table S5.

Added to the legend of Table S6:

For genes *BPGM* and *RUNX1* no eQTLs were found at 5% FDR (18).

5 Sensitivity analysis regarding pleiotropy

Results (Lines 133-140): This part is too cryptic. To allow better understanding please make sure that Readers can find the respective part in the Methods section by, for example, mention in Lines 371-376 the term “direct adjustment” or alternatively, the phrase “assessment of residual pleiotropy” in the Results part, or both in order to give some cross-reference.

We added additional information about the sensitivity analyses in the Methods sections to make this more clear.

Added to the Results:

As we previously showed, correction for this source of **direct** pleiotropy can be achieved by including the variant as a covariate in the MR model (4). **This adjustment** corroborated results of Cochran’s method for all identified effects and both adjustments negated the case of pleiotropy in which the variant rs174546 has a direct effect on *LPL* expression not mediated by TG level (P-value < 0.05; Figure S3).

Added to the Method section:

Sensitivity analyses were performed to assess residual pleiotropy using multivariable MR (20) with either 1) the dosage of the *cis*-expression QTLs in the GIVs as extra covariate in the MR model to adjust for direct pleiotropy, or 2) the potential pleiotropic GIV as extra covariate in the MR model to adjust for pleiotropy through a parallel path.

Please expand the Methods part (Lines 371-376) as I do not understand what the Authors have actually done, also with respect to Figure S3. From the presented formula I would assume that per transcription, several models were fitted (for each single variant) but this does not fit to the display in Figure S3. From that figure I can also not deduce the jump to rs174546 mentioned in Results (Line 139). Please also revise the legend in Figure S3 as the terms 'none', etc cannot be understood and explain what selection of results is actually displayed.

Added to the Methods:

Sensitivity analyses were performed to assess residual pleiotropy using multivariable MR (20) with either 1) the dosage of the *cis*-expression QTLs in the GIVs as extra covariate in the MR model to adjust for direct pleiotropy, or 2) the potential pleiotropic GIV as extra covariate in the MR model to adjust for pleiotropy through a parallel path.

Added to the legend of Figure S3:

Label “None” is the MR estimate without adjustment for pleiotropic effects, label “Q-stat” is the MR estimate adjusted for pleiotropic effects based on Cochran’s method, label “Local variant” is the MR estimate adjusted for direct pleiotropic effects by adding the dosage of the *cis*-expression QTL as covariate in the model.

In general, I would suggest to add all findings (incl. those from other sensitivity analyses) to Table S5 if not done already.

Yes, good suggestions. We added a column Sensitivity to Table S5 describing whether an effect was excluded based on several sensitivity analyses.

Added to the legend of Table S5:

Column Pleiotropy describes if and at what stage an effect was excluded based on several pleiotropy sensitivity analyses.

Results (Lines 145-150): There is no explanation given how GIVs for BMI, systolic and diastolic blood pressure were defined. Please add with everything that needs to be presented.

Added to the Results:

Finally, effect sizes were generally not sensitive to further adjustment for GIVs for the other lipids using a multivariable MR analysis (20) (e.g. the effect of TG on transcription was adjusted for GIVs of LDL-C and HDL-C etc.) and for the GIVs for potential

pleiotropic factors systolic blood pressure, diastolic blood pressure and BMI, which were constructed using public GWAS data (21,22) with the same procedure as for the lipid GIVs (Figure S4).

Added to the Methods:

GIVs for BMI, systolic blood pressure and diastolic blood pressure were constructed using public GWAS data (21,22) with the same procedure as for the lipid GIVs.

Results (Lines 151-155): This whole chapter ends with the statement that 2 genes are excluded from further consideration. Please make this clearer. I assume you mean the two genes LYBD6B and CCNA1 marked in Table S5. This might be achieved by also including consequences such as this to Table S5.

Added to the Results:

In summary, our systematic analysis of pleiotropy indicates that Cochran's method was generally successful in detecting and correcting for pleiotropy with the exception of 2 out of 62 effects (*LYBD6B* for TG and *CCNA1* for HDL-C). This resulted in a final set of 55 genes whose expression was influenced by TG and 5 genes whose expression was influenced by HDL-C (Table S5). As noted previously, HDL-C-influenced genes were also identified as TG-influenced genes. Our extensive analysis did not indicate pleiotropy and thus favors the interpretation that the effects on TG and HDL-C were independent.

6 Enrichment analysis

Methods (Lines 383-387): The Authors used Enrichr to assess pathway enrichment using 10 different databases. Since presented terms in Table S8 seem overlapping, the question arises why using all those 10 databases? How do they differ? Was the mapping done via gene name or Ensembl ID? Could everything mapped?

We changed the enrichment from Enrichr (which uses a black-box web API) to the enricher function of the clusterProfiler R package, using the same 10 databases. This has several advantages: 1) we can use a background of our expressed genes, 2) we can adjust for multiple testing over all databases at the same time, and 3) we can observe how many of the genes were mapped to the terms.

We decided to use multiple databases, since the combination alleviates some of the blind spots of the individual databases. The mapping was done based on gene symbols; of the 52 genes with a gene symbol (out of 55), 39 genes were present in at least 1 database.

Changed the Methods:

Pathway enrichment was performed using clusterProfiler (30) with a background set of 17,740 genes that were expressed in our data. The 10 human pathway databases BioPlanet 2019, WikiPathways 2019 Human, KEGG 2019 Human, Elsevier Pathway Collection, BioCarta 2015, Reactome 2016, HumanCyc 2016, NCI-Nature 2016, Panther 2016 and MSigDB Hallmark 2020 were queried using gene symbols, with 39 of 55 queried genes present in at least 1 database. Multiple testing using the Benjamini-

Hochberg method at 5% FDR was performed over the combined results from the 10 databases.

Results (Lines 163-189): Presentation of results are difficult to retrace only be having Table S8. I would thus suggest to expand Table S8 relating genes to lipid traits and terms (in text named as processes) to classifications (e.g., lipid metabolism). Any further support to deduce numbers stated in Results (e.g., 22 processes related to allergy, Line 175) can only be appreciated because of similarities of terms.

We see this point. We added the column to Table S8:

Column Type, which describes whether the pathway is part of lipid metabolism or allergy-related processes.

Figure 3B: If Table S8 is expanded it might also help to explain the grouping of genes not explained otherwise in the legend of this figure.

With the changes to Table S8 this should now be clear.

7 2-sample MR analysis related to allergy phenotypes

Methods (Lines 388-389): Please expand. Essentially everything is missing. Why those 4 phenotypes? What are these studies? How are traits defined? Selection of GIVs? ...

We agree with the Reviewer that this part of the Methods was short on details and we have revised the section accordingly.

Changed in the Methods:

Finally, we assessed the effects of genes affected by TG on allergy phenotypes and incidence of several chronic diseases using two-sample MR (57). We selected several large allergy GWAS studies for their precise estimates, i.e. a combined allergic disease phenotype (asthma and/or hay fever and/or eczema) (26), incidence of childhood-onset eczema (25) and incidence of adult-onset eczema (25), and a smaller but more specific GWAS on IgE levels (27). We also selected several large chronic disease incidence GWAS studies for their link with lipids and/or inflammation, namely coronary artery disease (28), myocardial infarction (28), and rheumatoid arthritis (29). We used for each gene the strongest associating *cis*-expression QTL (18) as a proxy for gene expression and used the same Wald method based MR approach as described previously.

8 Discussion

Overall, the Discussion is thorough. The Authors stated under limitations that relevant disease outcomes were not available. What disease outcomes do the Authors miss? Is this meant in reference to allergy phenotypes?

The Reviewer is correct that we meant allergy phenotypes. When we discovered the link between TG and allergy genes we would have liked to perform an analysis with allergy phenotypes in our own data instead of using public data.

Changed in the Discussion:

Limitations of our analysis include that it was based on whole blood **and that information on allergy-related outcomes was unavailable in our own data.**

Reviewer #2 (Remarks to the Author):

The authors present a novel approach to test for causality between biomarkers and transcription, and transcription and disease end points. The paper correlates standard lipid measurements with whole blood transcriptome of 3229 individuals and uses Mendelian randomization approaches to identify 55 genes where triglyceride levels potentially cause transcriptional changes, some with links to allergic response.

The paper is mostly clear and has potentially interesting results.

We thank the Reviewer for the valuable comments and we have revised the manuscript accordingly.

However, there are several points in analyses and interpretations of the results that would need clarification:

1. One of the cohorts LLS did not fast before lipid measurements. How different are the TG measures in LLS compared to other cohorts? It would be good to do sensitivity analysis of without LLS to make sure that the results are not driven by differences in fasting.

To address the Reviewer's concern, we now report the direction of effects for all cohorts separately in Table S1. Importantly, the results obtained from the non-fasting cohort LLS were in line with those from the other cohorts. For 100% (905/906) of genes, the

LLS cohort showed the same direction as the combined estimate of all cohorts. Overall, for 84% (757/906) of genes the direction was the same in all 6 cohorts indicating that our findings were independent of cohort-specific factors.

Added to the Results:

For 757 of 906 associations the direction was consistent for all 6 cohorts, for 143 associations the direction was consistent for 5 of 6 cohorts, for 5 associations the direction was consistent for 4 of 6 cohorts, and for 1 association the direction was consistent for 3 of 6 cohorts (Table S1).

2. There seemed to be very limited information on how the GIVs were constructed. How were the variants chosen for GIVs? How did you account for the LD between variants?

The variants included in the GIV were all variants reported to be genome-wide significant in a GWAS among 188,577 individuals and were >1 Mb apart with and $r^2 < 0.10$. We added this information to the Methods.

Added to the Methods:

GIVs were created for TG, HDL-C and LDL-C using **all** genetic variants reported in a GWAS among 188,577 individuals **that were >1 Mb apart and nearly independent ($r^2 < 0.10$)** (17) using the following equation:

3. The GIVs were tested against the potential confounders listed in Table S4. How were these confounders chosen? Lipid variants are known to associate with many other risk factors like BMI, WHR, blood pressure etc. and disease risks like CAD/stroke/T2D as well which may potentially also affect expression levels. These variables are also available in most if not all of the participating cohorts as well. These could also be tested in the cohorts or at least their effect could be discussed.

We agree that the specificity of the GIVs is an important point. We tested the association between GIVs and potential covariates that were unlikely to be a mediator in our analysis (age, sex, cell counts). We addressed the potential association of the GIVs with lipid-associated disease phenotypes like BMI and blood pressure as a potential cause for pleiotropy in the multivariable MR analysis.

Added to the Results:

Finally, effect sizes were generally not sensitive to further adjustment for GIVs for the other lipids using a multivariable MR analysis (20). **Similarly, we constructed GIVs for potential pleiotropic factors systolic blood pressure, diastolic blood pressure and BMI using public GWAS data (21,22) and the same procedure as implemented for the lipid GIVs (Figure S4). This analysis did not detect any further pleiotropic associations.**

4. The two sample MR analysis on expression levels and allergy could be benchmarked against similar analysis of TG-associated genes and CAD & T2D. As allergies also typically has onset in early life, it is also not quite clear how existing allergies may affect expression levels of the genes measured later in life as in these cohorts. This could be discussed as well.

We agree. Actually, we originally hypothesized our study would be relevant for cardiovascular outcomes and rheumatoid arthritis, both lipid-associated inflammatory diseases (first paragraph introduction) but had not followed up on this hypothesis. In line with the Reviewer's criticism we extended our 2 sample MR analysis. We chose not to report the result for type 2 diabetes, because this disease was not among our original hypothesis, but did do the analysis and observed an association for the TG-affected *CCR3* gene ($P\text{-value} = 2.4 \times 10^{-2}$), which however was no longer statistically significant after multiple testing correction ($P_{\text{FDR}} = 0.66$).

Added to results:

Finally, we performed the same two-sample MR analysis for atherosclerosis-related outcomes and rheumatoid arthritis, two lipid-associated inflammatory diseases for which we a priori hypothesized that lipid-induced changes in transcriptome of immune cells could be relevant. Based on summary statistics from GWAS of coronary artery disease (28), myocardial infarction (28), and rheumatoid arthritis (29), only for *CCR3* ($P\text{-value} = 6.0 \times 10^{-5}$) an effect on rheumatoid arthritis was indicated (Table S9). Several other TG-affected genes showed weaker associations with atherosclerosis-related outcomes (*ABCG1*, *AC004791.2*, *CPA3*, *CYP11A1*, *SLC12A3*) or rheumatoid arthritis (*GATA2*,

SMPDL3A) but were no longer statistically significant after correction for multiple testing ($P_{\text{FDR}} > 0.05$; Table S9).

Added to Discussion:

Our original hypothesis, however, was that the interaction between blood lipids and immune cells was particularly relevant for lipid-associated inflammatory diseases including atherosclerosis (5) and rheumatoid arthritis (6). With the exception of the *CCR3* gene for rheumatoid arthritis, none of the TG-affected genes were associated with these diseases on the basis of the two-sample MR approach. This negative result should be interpreted with caution in view of the limitations, including low statistical power, of the approach and the observation that some genes did show an association before the necessary correction for multiple testing. While any contribution is less apparent than for the allergy response, other study designs will be required to obtain a definite answer.

Added to Methods:

We also selected several large chronic disease incidence GWAS studies for their link with lipids and/or inflammation, namely coronary artery disease, myocardial infarction, and rheumatoid arthritis.

5. How was the considerable multiple testing burden controlled for in different stages of the analyses?

We have revised the manuscript in all relevant places to make this more clear. In short: the TWAS analysis was adjusted for 17,740 tests, i.e. the number of genes, using the Bonferroni method; the MR analyses for the number of TWAS associations per lipid, using the Benjamini-Hochberg method at 5% FDR; the enrichment analysis for the combined results from the 10 databases, using the Benjamini-Hochberg method at 5% FDR.

Changed sections in the Methods:

The results of each cohort were combined using a fixed-effect meta-analysis and P-values were adjusted for 17,740 tests, i.e. the number of genes, using the Bonferroni method.

Robust standard errors were calculated using Fieller's theorem (54) and P-values were adjusted for multiple testing, i.e. the number of TWAS associations per lipid, using the Benjamini-Hochberg method at 5% FDR.

Multiple testing using the Benjamini-Hochberg method at 5% FDR was performed over the combined results from the 10 databases.

Minor comments:

We thank the Reviewer for pointing out these errors.

- In Figure 1 Panel B, please fix the X-axis.

Fixed the X-axis of Figure 1B.

- In Discussion, line 230 please change “mutations” -> “variants”

This was based on Clee et al. [2] who investigated mutations in ABCA1 in families. We therefore kept this as “mutations”.

- Discussion, line 249, please add “potential” to “This study the first study with sufficient statistical power to find potential causal relationships...”

Added “potential” to line 249.

- In Discussion, line 271, please remove the work “rigorous”

Removed “rigorous” from line 271.

Reviewer #3 (Remarks to the Author):

In this study, Dekkers et al study the whole blood transcriptome of >3000 human subjects and use a mendelian randomization approach (more specifically GIV) to study which genes are transcribed under the influence of blood lipids through genetics. This is an elegant approach, allowing you to rule out potential environmental upregulation of lipid levels, and more importantly, study directionality. By doing this, they found 55 genes in immune cells to be genetically associated with triglycerides, which are genes in lipid metabolism pathways and allergic disease. The finding of genes in allergic disease are new and surprising, although the methods used (and more specifically the covariates added in the model) do rise some questions to the validity of these findings.

We thank the Reviewer for the valuable comments and we revised the manuscript accordingly. We discuss in detail the removal of basophils from the model and show with a new supplementary figure that this did not meaningfully affect the results.

Although the scope of my research is outside the use of MR methodology, I do think there are other important questions to be raised:

1. Why were 10 human pathway databases used to analyse the gene set? In fact, more pathways were found than genes. This feels quite like the fishing expedition. Did the authors correct for the multiple analyses that were performed?

We decided to use multiple databases, since the combination alleviates some of the blind spots of the individual databases. However, we now changed our analysis method from enrichR to clusterProfiler which allowed us to adjust for multiple testing over all the databases at the same time in line with the Reviewer's criticism.

Added to the Methods:

Pathway enrichment was performed using clusterProfiler (30) with a background set of 17,740 genes that were expressed in our data. The 10 human pathway databases BioPlanet 2019, WikiPathways 2019 Human, KEGG 2019 Human, Elsevier Pathway Collection, BioCarta 2015, Reactome 2016, HumanCyc 2016, NCI-Nature 2016, Panther 2016 and MSigDB Hallmark 2020 were queried using gene symbols, with 39 of 55 queried genes present in at least 1 database. Multiple testing using the Benjamini-Hochberg method at 5% FDR was performed over the combined results from the 10 databases.

2. The most striking and new result is the link between TG genotypes and genes in allergic disease. (It is quite obvious that genes in lipid metabolism are affected by TG) However, I am concerned that this is in fact an artifact of the covariates applied to (and/or left out of) the model. In the methods section, the authors describe that basophils were left out of the covariates that were corrected for due to the sum of 100%. This makes sense, but for sensitivity analysis (and the potential bias of the data), the same analysis should be repeated with eosinophils left out instead of basophils. Another option would be to work with numbers of

cells and not % (those can be calculated using the WBC and the % together).

Could the authors repeat the analysis with the raw numbers of cells and not the %? Are the genes related to basophils then still upregulated?

The rationale for adjusting for the number of cell count fractions minus 1 is that fractions add up to 1 so that including all fractions leads to collinearity in the model (and wrong effect estimates for the cell counts). When including 4 out of 5 cell counts, like we did, the effect of the 5th cell count will end up in the intercept. So, even though we left out 1 cell count, basophiles, we did correct for all cell counts. We agree that if we knew beforehand that basophiles were important, it would have been more logical to leave out another cell count. As indicated by the Reviewer we repeated our analyses including basophiles but now excluding eosinophiles. This analysis resulted in similar effect sizes illustrating in line with our expectation (see Figure S5).

Our models included total WBC count. In combination with the cell count fractions, this implies the model adjusts for absolute cell counts as individual fractions form the number of cells as a linear combinations.

Figure S5. Scatter plot of MR estimates with basophils removed and with eosinophils removed from the model.

Added to the Methods:

Since the sum of all cell type fractions was 100%, one of the cell types (here, basophils) was not included as covariate to prevent collinearity; the effect of the excluded cell type is captured in the intercept and thus corrected for implicitly. *Nota bene, omitting eosinophils instead of basophils did not meaningfully alter the results (Figure S5).*

3. Was sensitivity analysis performed on the origin of the data? Some cohorts described are general public cohorts, whereas other cohorts (especially the CVD/Diabetes cohort) are more specific. Did some cohorts skew the results? The same question arises for the fasting / non-fasting analysis of lipids

We added information about the direction of association of the cohorts to Table S1. We observed that out of 906 associations, CODAM (49/906) and PAN (63/906) are most often inconsistent with the direction of the combined estimate of all cohorts. These are also the smallest cohorts. LLS (the only cohort with non-fasted samples) was only inconsistent for 1 out of 906 associations. All cohorts with fasted samples are either general population cohorts (LL, RS), twins (NTR), healthy controls (PAN), or included participants based on a moderately increased risk of developing cardiometabolic diseases, such as type 2 diabetes and/or cardiovascular disease (CODAM). We do not see that any of these drive the

Added to the Results:

Evidence for the associations was generally consistent across cohorts: for 757 of 906 associations the direction was consistent for all 6 cohorts, for 143 associations the direction was consistent for 5 of 6 cohorts, for 5 associations the direction was consistent for 4 of 6 cohorts, and for 1 association the direction was consistent for 3 of 6 cohorts (Table S1).

Added to Table S1 the column Direction containing the direction of associations for the cohorts.

Added to the legend of Table S1:

Column Direction contains the direction of association for cohorts CODAM, LL, LLS, NTR and PAN, respectively.

4. The authors start the abstract and introduction with an interest in priming of immune cells by lipids and trained immunity. However, by using an MR approach, the authors actually rule out the effects of lipids and environmental / temporary increase by using the genetic approach. All immune cells will also intrinsically be affected by these genetic variations which could lead to the changes in gene expression rather than due to the rise in TG levels in the circulation. Furthermore, the authors do not discuss this phenomenon of trained immunity or priming at all anymore in the rest of the paper. Do the authors believe they have added to the understanding of trained immunity by lipids? If not, it should be removed from the abstract and introduction as it is misleading.

The Reviewer is correct. We removed all references to priming and trained immunity from the manuscript. Thanks for pointing this out.

5. Following up on the fourth point: the 55 genes that were found to be influenced by genetics are only a small part of the 496 initially associated TG genes. Does this mean that the other genes are more environmentally associated? That could be an interesting point of view, as environmentally associated genes are

potentially more reversible than genetics. Did the authors look at the difference between the two findings and what pathways are associated with those genes?

We believe the reduction from 496 genes in the observational analysis to 55 genes in the MR analysis mainly stems from the large difference in statistical power between the methods. The post-hoc power analysis we included in the manuscript makes this point. As we note in the Discussion: ‘a study size larger than a million individuals is required to have sufficient statistical power to detect 90% of the effects.’ This is also apparent from Figure 2A, which plots the effects sizes in the observational study versus the MR analysis. For many genes, the effect sizes are comparable but only for the largest effect sizes, the MR analysis is significant. Of course, there also are MR estimates of 0. It remains speculative if this is due to a larger environmental influence on gene expression or the presence of confounding in the observational analysis that was solved in the MR analysis. Therefore, we are unsure how to interpret genes that do not survive triangulation, i.e. being detected using different methodology, and prefer not to add such analysis including a discussion of its limitations.

Minor comments

We thank the Reviewer for pointing out these errors.

Line 172-175: The authors mention three genes, after which two genes are actually explained. What is the third additional gene?

Changed to 2 genes.

Line 253: This study the first study – “is” is missing

Added “is” to line 253.

Figure 1b: the x-axis font size is too big: words are overlapping

Fixed the X-axis of Figure 1B.

References

- [1] U. Võsa *et al.*, “Unraveling the polygenic architecture of complex traits using blood eQTL metaanalysis,” *bioRxiv*, p. 447367, Oct. 2018, doi: 10.1101/447367.
- [2] S. M. Clee *et al.*, “Age and residual cholesterol efflux affect HDL cholesterol levels and coronary artery disease in *ABCA1* heterozygotes,” *J Clin Invest*, vol. 106, no. 10, pp. 1263–1270, Nov. 2000, doi: 10.1172/JCI10727.

REVIEWERS' COMMENTS

Reviewer #1 (Remarks to the Author):

Thank you for the revision of the manuscript. The Authors sufficiently addressed my comments. Overall, the manuscript has greatly improved. I have nothing to add.

Reviewer #2 (Remarks to the Author):

The authors have done a good job in revising the manuscript and answering to the requests.

Reviewer #3 (Remarks to the Author):

I am happy with the revisions and answers to my questions and I have no further comments.

REVIEWERS' COMMENTS

Reviewer #1 (Remarks to the Author):

Thank you for the revision of the manuscript. The Authors sufficiently addressed my comments. Overall, the manuscript has greatly improved. I have nothing to add.

Reviewer #2 (Remarks to the Author):

The authors have done a good job in revising the manuscript and answering to the requests.

Reviewer #3 (Remarks to the Author):

I am happy with the revisions and answers to my questions and I have no further comments.

We are happy that the reviewers have no further comments.